

# Planktonic foraminiferal spine versus shell carbonate Na incorporation in relation to salinity

Eveline M. Mezger[1], Lennart J. de Nooijer[1], Jacqueline Bertlich[2], Jelle Bijma[3], Dirk Nürnberg[2], Gert-Jan Reichart[1,4]

[1]Department of Ocean System Sciences, Royal Netherlands Institute for Sea Research, and Utrecht University, Texel, Netherlands

[2]GEOMAR Helmholtz Centre for Ocean Research Kiel, Wischhofstr. 1-3, 24148 Kiel, Germany

[3]Alfred-Wegener-Institut Helmholtz-Zentrum für Polar- und Meeresforschung, Am Handelshafen 12, Bremerhaven 27570, Germany

[4]Department of Earth Sciences, Faculty of Geosciences, Utrecht University, Heidelberglaan 2, 3584 CS Utrecht, Netherlands

*Correspondence to*: Eveline M. Mezger (Eveline.Mezger@nioz.nl)

**Abstract.** Sea surface salinity is one of the most important parameters to reconstruct in paleoclimatology, reflecting amongst others the hydrological cycle, paleo-density, ice volume, and regional and global circulation of water masses. Recent culture studies and a Red Sea field study revealed a significant positive relation between salinity and Na incorporation within benthic

and planktonic foraminiferal shells. However, these studies reported varying partitioning of Na between and within the same species. The latter could be associated with ontogenetic variations, most likely spine loss. Varying Na concentrations were observed in different parts of foraminiferal shells, with especially spines and regions close to the primary organic sheet being enriched in Na. In this study, we unravel the Na composition of different components of the planktonic foraminiferal shell wall using Electron Probe Micro Analysis (EPMA) and solution-ICP-MS. A model is presented to interpret EPMA data for spines and

spine bases to quantitatively assess differences in composition and contribution to whole shell Na/Ca signals. The same model can also be applied to other spatial inhomogeneities observed in foraminiferal shell chemistry, like elemental (e.g. Mg, Na, S) banding and/or hotspots. The relative contribution of shell calcite, organic linings, spines and spine bases to whole shell Na chemistry is considered quantitatively. This study shows that whereas the high Na areas may be susceptible to taphonomy, the Na chemistry of the shell itself seems relatively robust. Comparing both shell and spine Na/Ca values with salinity shows that

shell chemistry records salinity, albeit with a very modest slope.



## 1 Introduction

Salinity is one of the most wanted parameters to reconstruct in paleoceanography, driving together with temperature, the

thermohaline circulation as well as reflecting regional hydrological cycling. Whereas temperature can be reconstructed by a variety of proxies (e.g. $U^{k'}_{37}$ (Prahl and Wakeham, 1987); foraminiferal Mg/Ca (Elderfield and Ganssen, 2000;Lea et al., 1999;Nürnberg et al., 1996); foraminiferal $\delta^{18}O$ (e.g.(Zachos et al., 2001;Elderfield and Ganssen, 2000)) and $TEX_{86}$ (Schouten et al., 2002), equally reliable proxies for salinity are still under development (Wit et al., 2013b;Mezger et al., 2016;Allen et al., 2016;Rohling and Bigg, 1998;Schouten et al., 2006). A number of approaches have been proposed to reconstruct salinity,

including a combination of stable isotopes ($\delta^{18}O$ from foraminiferal shells or $\delta D$ of long chain ketones) with independent reconstructions of sea surface temperature (e.g. Mg/Ca or $U^{k'}_{37}$, (Elderfield and Ganssen, 2000;Schouten et al., 2006)); foraminiferal Ba/Ca (Weldeab, 2007), dinoflagellate cyst morphology (e.g. (Verleye et al., 2012;Mertens et al., 2012)) and $\delta D$ of long chain ketones and alkenones (e.g. (Vasiliev et al., 2017)). However, uncertainties associated with the indirect controls on these proxy signals or preservation issues result in (large) errors in the reconstructed salinity (Rohling, 2007). This can be

circumvented by a more direct approach, related to elements determining seawater salinity (e.g. Cl, Na). Even though Na is considered as a conservative element in seawater, recent culture studies and a Red Sea field study reveal a significant positive relation between salinity and Na incorporation within benthic (Wit et al., 2013b;Geerken et al., 2018) and planktonic (Allen et al., 2016;Mezger et al., 2016;Bertlich et al., 2018) foraminiferal shells. This relation between salinity and Na incorporation, potentially related to an increase of the $Na^+/Ca^{2+}$ activity ratio with salinity, is not only observed for foraminiferal calcite (Allen

et al., 2016;Mezger et al., 2016;Wit et al., 2013b), but also for barnacles and Atlantic oyster shells (Rucker and Valentine, 1961;Gordon et al., 1970) and inorganically precipitated calcium carbonate (Kitano et al., 1975;Ishikawa and Ichikuni, 1984).

Previous studies on the incorporation of Na into biogenic and inorganic calcite varied in reported partition coefficients, despite similar conditions (White, 1978;Ishikawa and Ichikuni, 1984;Kitano et al., 1975). These differences are not only observed between inorganic and biogenic studies, but also between and within the same foraminiferal species, either growing in

culture or the natural environment (Mezger et al., 2016;Allen et al., 2016;Wit et al., 2013b;Bertlich et al., 2018). Recently (Mezger et al., 2018) studied the preservation of the Na-salinity signal of the *G. ruber* and *T. sacculifer* species through the water column, comparing sedimentary and water-column collected specimens (0-500 m) of the Red Sea. It was found that Na/Ca values decrease with water depth, thereby aligning the lower Na/Ca from the surface sediment samples with those observed in culture studies (Allen et al., 2016;Wit et al., 2013b;Mezger et al., 2018;Bertlich et al., 2018). The loss of spines, highly enriched

in Na (Jacob et al., 2017;Branson et al., 2016;Mezger et al., 2016), during settling in the water column is hypothesized to be the controlling factor of the decreasing Na/Ca values, as foraminifera shed their spines before gametogenesis (Bé, 1980;Zhao et al., 2017). Furthermore, it has been suggested that calcite growth rate (Busenberg and Plummer, 1985), temperature (Allen et al., 2016), environmental differences between field and controlled growth experiments (Wit et al., 2013b;Allen et al., 2016;Mezger et al., 2016), life stages (Mezger et al., 2018)ageing/leakage (Yoshimura et al., 2017), or organic linings (Yoshimura et al.,

2017;Branson et al., 2016) affect Na incorporation. The inhomogeneous inter-shell distribution of sodium, partially due to life stage, could influence measured Na values (Geerken et al., 2018;Branson et al., 2016;Mezger et al., 2018), and potentially explain part of the observed differences. Similar to reports for other inter-shell element distributions (e.g. Mg; (Sadekov et al., 2005;Hathorne et al., 2009;Kunioka, 2006)), Na appears to occur in bands of alternating high- and low concentrations (Geerken et al., 2018). However, the thickness and intensity of these bands is not similar between species (Geerken et al., 2018). For the

planktonic species *Globigerinoides ruber* and *Trilobatus sacculifer*, elevated concentrations of Na are also observed in regions where the spines meet the rest of the shell wall (Branson et al., 2016;Mezger et al., 2018), close to the Primary Organic Sheet (POS). This may indicate that different species vary in their calcification mechanisms: i.e. spines and gametogenic (GAM)



calcite in planktonic species may be precipitated by different biomineralization pathways and hence, may have various element compositions (Steinhardt et al., 2015;Nürnberg et al., 1996;Sadekov et al., 2005). Clearly, the internal Na distribution influences

measured Na/Ca values and is hence important for the potential application of foraminiferal shell Na/Ca for salinity reconstructions. In this study, different parts of planktonic foraminiferal shells are distinguished geochemically to quantify the relative contribution of shell calcite, spine calcite and organic linings on the total foraminiferal Na/Ca. We use high resolution quantitative Electron Probe Micro Analyses (EPMA) to distinguish differences in element composition between different parts of the shell and Scanning Electron Microscopy (SEM) to determine the relative contribution of spines (thickness, density) of

surface water specimens. Not only field collected surface water specimens (Mezger et al., 2016), but also cultured *T. sacculifer* (Bertlich et al., 2018), Red Sea water column and surface sediment *T. sacculifer* and *G. ruber* specimens (Mezger et al., 2018) are measured for comparison. Furthermore, we assess the impact on the foraminiferal shell chemistry of the organic linings by isolating these linings and analyzing their Na/Ca. These data are subsequently evaluated along a (surface water) salinity gradient, considering the potential impact of taphonomy to evaluate the proxy potential of foraminiferal shell Na/Ca values.

**2 Methods**

Living Red Sea field-collected *T. sacculifer* and *G. ruber* specimens were collected in May 2000 during R/V Pelagia cruise 64PE158 (Mezger et al., 2016). Core-top and box-core (upper 0-1 cm) specimens from similar locations were collected during different cruises as described in (Siccha et al., 2009). Cultured *T. sacculifer* specimens were collected at 3−8 m water depth 1−2 miles off the south coast of Curacao and off the west coast of Barbados, after which they were grown in filtered

seawater with salinities ranging from 26 to 45 (Nürnberg et al., 1996;Bijma et al., 1990;Bertlich et al., 2018). To study the relative contribution of Na in different parts of the shell to the total Na/Ca composition, high resolution quantitative Electron Probe Micro Analyses was performed at Utrecht University (section 2.1). Spine thickness and densities (number of spines per surface area) were derived by Scanning Electron Microscopy (section 2.2). For the chemical analyses of organic linings (section 2.3), foraminifera within the size fraction of 250-355 µm for *T. sacculifer* and 100-355 µm for *G. ruber* were collected from

calcareous ooze, retrieved by a gravity core at the Walvis ridge (similar to the material used for the NFHS-1, (Mezger et al., 2016)).

**2.1 Electron Probe Micro Analyses (EPMA)**

The Na/Ca composition of the spines and shells collected from the Red Sea water column, and core-tops were measured at a high spatial resolution using EPMA (Table 1). Several specimens of both species were selected and embedded in resin

(Araldite 20/20) in a vacuum chamber. Multi-net collected specimens were isolated directly upon low temperature ashing for spine analysis and transferred without sieving to preserve the spines and embedded in resin as well (Mezger et al., 2018). After drying for at least 48 hours in an oven at approximately 50°C, the specimens embedded in resin were polished. Upon polishing samples were cleaned with double de-ionized water and coated with carbon after drying. Element mapping for Na and Ca of cultured specimens of *T. sacculifer* (Bertlich et al., 2018) as well as Red Sea-derived specimens of *G. ruber* and *T. sacculifer* was

performed with an electron microprobe at Utrecht University (JEOL JXA-8530F Field emission Electron Probe Micro Analyzer). Maps were generated with a focused electron beam, a beam current of 10 nA and an accelerating voltage of 7 kV. The dwell time was set at 300 ms and pixel sizes ranged between 0.2 and 0.43 µm. Counts, representing current strength, were converted to elemental ratios using analyses on standard material. We used Jadeite for Na, foraminiferal calcite for Ca and Forsterite for Mg, assuming a linear dependency of concentration (in mass %) on the signal and a constant background. Background intensities,





measured for the same (foraminiferal) samples with similar settings, were subtracted from total element intensities before converting to mass %. Single points were eliminated from further analyses when the Ca mass percentage of that spot was below 30%.

In this article, we refer to 'whole shell' for total shell measurements including high Na regions such as spines (e.g. laser ablation measurements in (Mezger et al., 2016)), and 'shell-only' when spine (base) regions are excluded from analysis. For the

elemental analyses of the foraminiferal shell, regions of the shell not containing spines (shell-only) were selected including potential banding, but excluding Na hotspots, which were observed near spine bases (e.g. Fig. 1 and Fig. S1). Deconvolving the "true" maximum Na values within the mixed spine signal is challenging, as the EPMA Na/Ca signal has a limited resolution. Values hence consist of shell calcite values, as well as pixel averaged mixed signals and the real spine signal. Because of the limited size of the spines and spine bases only a few of the analyses will capture spine carbonate, while more analyses capture a

mixture of both spine and ontogenetic carbonate and most analyses will show ontogenetic carbonate only. In the discussion, we suggest how the limited data of the spine chemistry can still be interpreted (section 4.2).

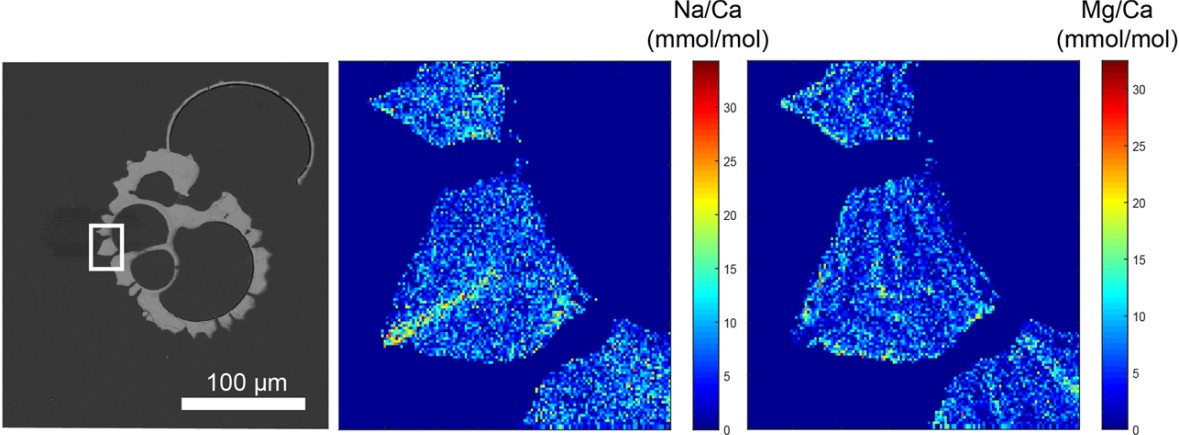

**Figure 1: Example of a backscattered electron overview image of an embedded and polished *G. ruber* plankton tow specimen with the white square indicating the zoom-in region (left), and the Na/Ca (middle) and Mg/Ca (right) EPMA images of this zoom-in, showing**
**the relative distribution of both elements within the shell. Whereas Na appears to be mainly concentrated in the spine (base), Mg mainly occurs in clear bands sloping upwards towards the spine and somewhat lower in the spine region (Mezger et al., 2018).**

The similarity between Na/Ca EPMA measurements of the same cultured *T. sacculifer* specimens performed at GEOMAR in Kiel, grown at different salinity and temperature conditions (for standards and measurements: (Bertlich et al., 2018)) and Utrecht University was used to assess consistency of the measurements, which was between 101.8% and 106.4% for

the line scans and between 101 and 122% for the maps (concentrations Utrecht/Kiel). These values are a conservative estimate, as the selection of the lines and regions to compare are never identical to previous measurements on the same shell (Fig. S2). Details for these cultured specimens can be found in (Bijma et al., 1990). Elemental analysis on JCP-1 powder (n=6) (Okai et al., 2002) were included to assess accuracy (sample/reference) of Sr (99.3%), Mg (106.3%), S (103.4%) and Na (85%). Although the error on the Na quantification is considerable, offsets are minor compared to the ranges studied here.

**2.2 Scanning Electron Microscopy**

Surface structures of foraminifera, including spine density, length and width, were quantified using scanning electron microscopy (SEM3000, Hitachi). However, as a consequence of sampling (plankton pump, sieving) and sample preparation (low temperature ashing – LTA (Mezger et al., 2016;Fallet et al., 2010)), many of the spines (partially) broke off and the total spine



lengths could not be determined and not used for further calculations. Spine density was calculated from pictures of a 50 μm x 50

μm square, focusing on the F-2 and if not available, the final or penultimate chamber. We used the surface water collected specimens for two species (*G. ruber* and *T. sacculifer*), which were measured previously for their Na/Ca composition with laser ablation-ICP-Q-MS (Mezger et al., 2016). Previously ablated areas were avoided, but using these exact specimens allows comparing the earlier published whole shell data (respectively shells including spine(s) (bases)) with the here presented spine distributions. The number of spines was determined by counting the number of pores, as these morphological features are more

robust. This quantification is based on the assumption that a spine is/was present at every corner of the cancellated (hexagonal) shell structure around each pore for these species (Bé, 1980). The thickness of the round spines was measured at the base of the spine. This effectively avoids potential pitfalls of the method associated with tapering of spines (Fig. 2). Foraminiferal size was measured as described by (Mezger et al., 2016).

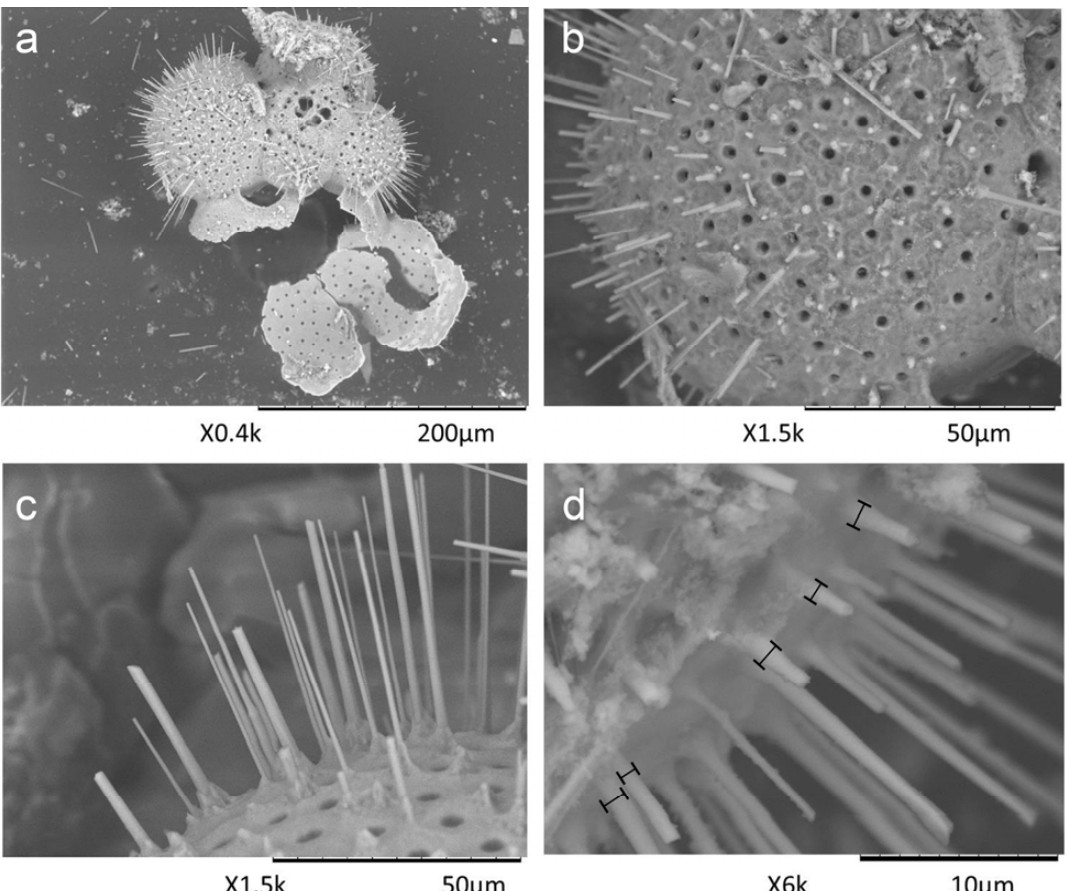

**Figure 2: Example of SEM-images of the foraminiferal specimens studied here: a) example of a laser-ablated *T. sacculifer* specimen, b) spine count area in the F-2 chamber, c) zoom-in of spines, showing the tapering shape of spines and d) spine width measurements at the base of the spines.**

## 2.3 Organic linings

For isolation of organic linings, 257 *G. ruber* specimens and 150 *T. sacculifer* specimens were selected from a

calcareous ooze isolated from a gravity core recovered from the Walvis ridge (similar to the material used for the NFHS-1, (Mezger et al., 2016)). After isolation of the specimens, samples were treated to remove organic matter on the outside of the shell



with buffered 1% hydrogen peroxide in a heated water bath at 90°C. Thereafter, samples were crushed lightly to enable clay particle removal from the inside of the shell by ultra-sonication. The calcite shells were subsequently dissolved in a glass beaker filled with 0.1M ultrapure weak acetic acid, leaving these overnight to dissolve. After visual inspection of the dissolution of the

shells, organic linings (OL) were isolated, centrifuged and rinsed three times with ultrapure water. Weights were determined after drying the isolated linings in a laminar cabinet at room temperature. In total, 0.04 mg OL was isolated from 5.66 mg *T. sacculifer* shells (0.7%), and 0.02 mg OL from 4.95 mg *G. ruber* (0.4%). After drying and weighing, the isolated OL was destructed in a PTFE tube overnight in an oxidative acid mixture (0.09 mL ultrapure $HNO_3$ and 0.01 mL ultrapure perchloric acid) in a 70°C water bath. The sample was brought to near dryness before being transferred to a PTFE digestion tube with 0.075

mL ultrapure $HNO_3$, and kept at 150°C for 12 hours. After cooling down another aliquot of 0.05 mL ultrapure perchloric acid was added and left to react at 180°C on a thermostatic block. After the samples were inspected for total destruction the sample was diluted to 2.5 mL with ultrapure water and small amounts of ultrapure $HNO_3$. The elemental composition was subsequently measured with a Thermo Fisher Scientific iCAP-Q. Elements were quantified using their relevant isotopes (respectively [23]Na, [24]Mg, [43]Ca and [88]Sr). Calibration standards used were taken up in a similar matrix (1M $HNO_3$). OL quantifications were based on

back calculating original shell and OL weights.

## 3. Results

### 3.1 EPMA

#### 3.1.1 Shell Na/Ca

Generally Na is rather homogeneously distributed throughout the shell, although Na hotspots are observed in spines and near the

spine bases (Fig. 1, Fig. S1). For none of the specimens from plankton pumps, core-tops or multi-nets, banding is observed, except for one specimen of *T. sacculifer* (Supplement part 3, pecimen 31-4). Several areas from shell cross-sections were selected in such a way to avoid areas enriched in Na ('shell-only'). This basically excludes areas with spines and spine bases. Plankton pump shell-only *G. ruber* Na/Ca values range from 5.6 ±0.18 to 7.7 ±0.25 mmol/mol (averages and standard errors) for a Red Sea surface water salinity of 37.3 and between 5.91 ±0.21 and 6.39 ±0.29 for a Red Sea surface water salinity of 39.6

(Table 2). For plankton pump collected *T. sacculifer*, shell-only Na/Ca values range between 6.12 ±0.20 and 6.83 ±0.13 mmol/mol for a Red Sea surface water salinity of 37.3 and between 6.12 ±0.15 and 6.75 ±0.31 for a Red Sea surface water salinity of 39.6 (Table 2). Shells collected from the 0 to 100 m water depth interval show Na/Ca values for *T. sacculifer* ranging from 5.6 ±0.12 mmol/mol to 7.1 ±0.10 mmol/mol and for *G. ruber* between 5.95 ±0.13 and 8.42 ±0.18 (Table 3). Core-top shell-only Na/Ca values range from 5.41 ±0.17 to 6.84 ±0.25 mmol/mol for *G.* ruber and from 5.52±0.14 to 6.22±0.23 mmol/mol for

*T. sacculifer* (Table 2). All quantitative EPMA maps for Na in the shells and spines are shown in the Supplement section.

#### 3.1.2 Spine Na/Ca

For the multi-net derived samples we were able to directly measure spine Na/Ca values on spines sticking out of the shell. Within the spines a considerable variability in Na/Ca values is observed, but not with a consistent zonation or trend. Spine *G. ruber* Na/Ca values for the multi-nets (S=~39.8) range from 10 ±1.3 mmol/mol to 23.5 ±1.9 mmol/mol, whereas *T. sacculifer*

Na/Ca values range from 10.7 ±0.8 mmol/mol to 24.9 ±1.9 mmol/mol. Intra-specimen spine variability is more than 200% for both *G. ruber* and *T. sacculifer* (highest/lowest average spine Na/Ca values, Tables 4, 5). Spine Na/Ca values are consistently much higher compared to shell Na/Ca values (e.g. Fig. 3, Tables 2-5). Comparing shell-based Na/Ca values with the Na/Ca





values measured on spines for the same specimen, for *T. sacculifer* spines are 2 to 4.3 times higher and for *G. ruber* spines are 1.4 to 2.5 times higher (Tables 2-5). No correlation is observed between spine and shell-based Na/Ca values for neither species.

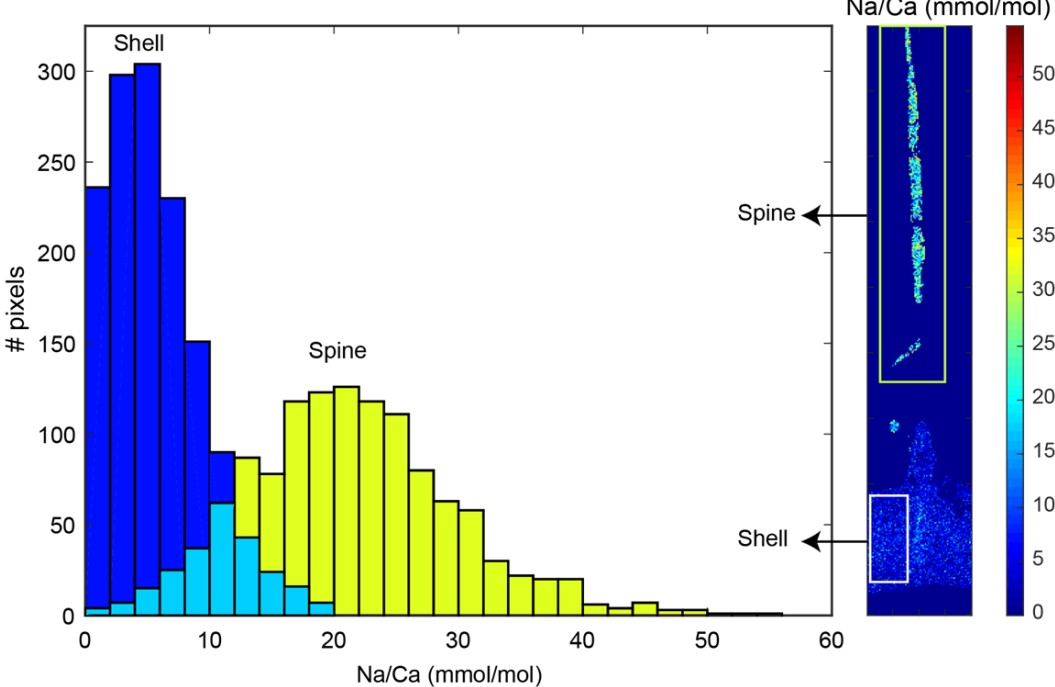


**Figure 3: Comparison between spine Na/Ca values (green columns) and shell Na/Ca values (blue columns) within the same specimen (specimen 0002_13, T. sacculifer, Table 3) and EPMA-map indicating the regions represented by the histograms (white box: shell, green box: spine). The turquoise color represents the overlap region of the spine and shell histogram (not the spine base). Clearly, spine Na/Ca values are higher compared to shell Na/Ca values.**

195       For several specimens we were able to measure both spines and spine bases. The EPMA-analyses show a clear difference in Na/Ca values between spines and spine bases. The spine bases show values in between the high spine and low shell Na/Ca values. Still, as the spine bases are surrounded by foraminiferal shell calcite, this possibly results in mixing signals between spine base and shell carbonate, which would decrease values for the spine base. Clearly, Na/Ca values will also depend on the selected cross section analyzed with EPMA. The distribution plots for the spine base Na/Ca values show a clear difference

from the shell-only areas and generally higher values (Supplement section 2 and 3).

## 3.2 Scanning electron microscopy

In total, 125 *G. ruber* and 38 *T. sacculifer* specimens were analyzed for their spine widths and spine density at the shell surfaces. In general, the number of spines is higher for *G. ruber* compared to *T. sacculifer*, whereas spine thickness is lower (Fig. 4). Spine density varied from 0.09 to 0.30 spines per $\mu m^2$ for *T. sacculifer* and from 0.12 to 0.30 spines per $\mu m^2$ for *G. ruber* (Fig. 4).

Spine widths show a high variability between and within specimens, ranging from 0.89 μm to 3.96 μm for *T. sacculifer* and from 0.56 um to 3.78 μm for *G. ruber*. A weak positive correlation is observed for *G. ruber* and *T. sacculifer* between spine width and the size of the foraminiferal shell (*G. ruber*: $R^2$=0.04, p<0.0001; *T. sacculifer*: $R^2$=0.04, p=0.004, based on Shapiro-Wilk test, Fig. 4). Salinity correlates negatively with spine width, based on weighted averages of the widths per salinity group for both





species (*G. ruber*: $R^2 = 0.35$, p<0.0001; *T. sacculifer*: $R^2 = 0.46$, p<0.0001) (Fig. 4). Furthermore a negative correlation is

observed between Na/Ca values and spine width (*G. ruber*: $R^2 = 0.016$, p=0.006; *T. sacculifer* : $R^2 = 0.03$, p=0.006 for).

A significant negative correlation is observed between foraminiferal shell size and the number of spines for both species (*G. ruber*: $R^2 = 0.17$, p<0.0001; *T. sacculifer* : $R^2 = 0.38$, p<0.0001, Fig. 4). Between salinity and spine density both species show a negative significant correlation (*G. ruber*: $R^2 = 0.24$, p<0.0001; *T. sacculifer*: $R^2 = 0.18$, p = 0.006, Fig. 4). However, average spine density values for *G. ruber* are not statistically different for the different salinities and therefore no correlation is observed

between salinity and spine density (student t-test between data points (p>0.78 for *G. ruber*)). For *T. sacculifer*, in contrast, spine density values differ statistically significant for the highest salinity compared to the other salinities (40.1, p<0.0375). The lowest salinity for *T. sacculifer* could not be taken into account for these calculations, because it only consisted of one single data point. No significant correlation is found between Na/Ca and spine density (*G. ruber*: $R^2 = 0.02$, p=0.1; *T. sacculifer*: $R^2 = 0.004$, p = 0.7).

### 3.3 Organic linings

The minor and trace elemental composition of the isolated organic linings is similar for *T. sacculifer* and *G. ruber* (Table 7). Although concentrations of Na and Mg seem enriched within the isolated organics (Table 7), when calculating their contribution to the whole shell elemental composition, Na from the linings contributes only 5.61 ppm, or 0.024 mmol/mol for *G. ruber* and 12.03 ppm, or 0.052 mmol/mol, for *T. sacculifer* to the whole shell Na/Ca. For Mg values the contribution from the

isolated linings to the total shell concentration are 37.68 ppm, or 0.16 mmol/mol for *G. ruber* and 69.25 ppm, or 0.28 mmol/mol for *T. sacculifer*. There is no measurable Ca concentration in the organic linings.



**Figure 4: Results from the SEM spine density and width counts for *G. ruber* (red closed circles) and *T. sacculifer* (blue open circles) specimens, with standard errors indicated and enveloped into a 95% certainty interval, with: a) number of spines versus the size (μm) of the foraminiferal specimen these were counted on, b) spine width (μm) versus the size (μm) of the specimen these were counted on, c) number of spines versus salinity, d) spine width versus salinity, e) number of spines and the measured whole-shell Na/Ca composition and f) spine width versus the measured whole-shell Na/Ca composition.**






## 4. Discussion

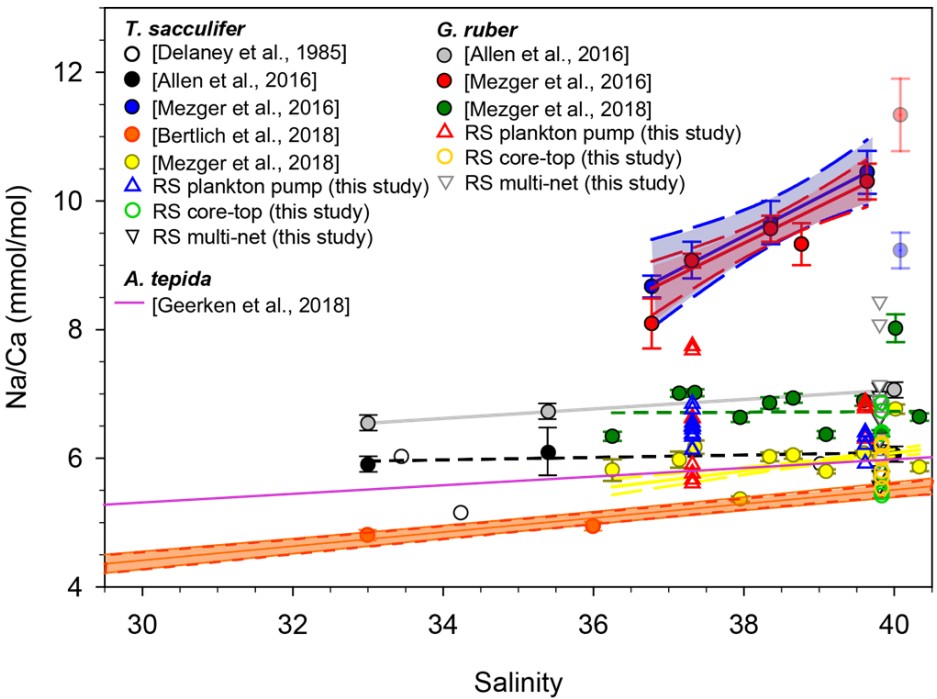

**Figure 5: Comparison of most existing planktonic foraminifer Na/Ca-salinity studies, including different culture studies (Allen et al., 2016, Bertlich et al., 2018, Wit et al., 2013, , Delaney et al., 1985), a field survey (Mezger et al., 2016, Mezger et al., 2018), and a benthic culture study (Geerken et al., 2018) compared to new electron microprobe shell Na/Ca values of Red Sea plankton pumps, core-tops and multi-nets (this study).**


Shell-only (i.e. spine-free) Na/Ca values of *G. ruber* and *T. sacculifer*, collected in the Red Sea from surface waters, the water column and the sediment surface, all fall within the range of previously established calibrations (Geerken et al., 2018;Allen et al., 2016;Wit et al., 2013b) (Fig. 5). Average values reported here are, however, somewhat higher than results from (Bertlich et al.,

2018). Red Sea sediment surface Na/Ca values measured by laser ablation (LA)-ICP-MS and EPMA from the same specimens compare well ((Mezger et al., 2018), Fig. 5). However, for the specimens collected from the sea surface, the EPMA-derived shell-only Na/Ca values are much lower than those from the LA-ICP-MS analyses (Fig. 5). When measuring whole-shell chemistry using LA-ICP-MS, all different shell components contribute to the signal including the Na-rich spines and spine bases. It is therefore hypothesized that spine loss is responsible for the observed offset in absolute Na/Ca between specimens from

surface water and those from deeper in the water column (Mezger et al., 2018). The fact that the shell-only Na/Ca values between core-top and surface water specimens are similar confirms this hypothesis (Fig. 5).

Several recent studies showed foraminiferal Na/Ca values to vary with salinity (Geerken et al., 2018;Wit et al., 2013b;Allen et al., 2016;Mezger et al., 2016;Mezger et al., 2018;Bertlich et al., 2018) (Fig. 5). The slopes of these calibrations and absolute Na/Ca values, however, differ between studies (Geerken et al., 2018;Allen et al., 2016;Wit et al., 2013b;Delaney et

al., 1985;Mezger et al., 2016;Mezger et al., 2018;Bertlich et al., 2018). Whereas some offsets may be due to inter-species differences, other offsets reflect variability within one species. Studies focusing on intra-shell variability in Na/Ca have shown that there are also large differences in Na/Ca within single shells (Branson et al., 2016;Geerken et al., 2018;Yoshimura et al., 2017;Mezger et al., 2018). Therefore, part of all this variability may be explained by uneven contributions of various parts of the foraminiferal shell, which means that the Na/Ca composition of these parts needs to be determined independently. This allows



calculating relative contributions of different parts of the shell to the whole-shell Na/Ca composition from previous studies. Based on suggestions made before, we here focus on the contribution of organic linings, spines and spine bases to the overall Na-composition of the foraminiferal shell.

### 4.1 Organic Linings

Using TOF-SIMS and an atom-probe, (Branson et al., 2016) found Na concentrations to be enriched at foraminiferal
spine bases and in (the proximity of) the organic linings. Values in the enriched areas appear approximately 1.3 times higher compared to the shell in *Orbulina universa* (Branson et al., 2016). (Geerken et al., 2018) discovered Na to preferentially occur in bands with concentrations 1.1-1.75 times higher compared to the surrounding layers with lower Na/Ca values. These bands seem to coincide with Mg-rich bands, which have previously been linked to the proximity of organic linings (Branson et al., 2016;Geerken et al., 2018). Although this coincidence suggests that high Na is indeed linked to the organic calcifying matrix, the
values measured on the isolated linings also indicate that their relative contribution to the overall shell Na/Ca is negligible (Table 7). Still, the higher concentrations of these linings might explain (part of the) observed banding pattern, as the absolute concentration within the linings is similar to or higher than that of the shell carbonate. One potential pitfall of the method used here for isolating the organic linings is that minor and trace metals adsorbed and/or loosely bound to the organic linings could have been removed during the rinsing phases of the isolation.

When not directly related to the organic layers, zones of high Mg and high Na may be indirectly coupled via processes responsible for the banding. For planktonic species, chamber formation (usually at night) may be responsible for the observed banding (Fehrenbacher et al., 2017;Spero et al., 2015). Here banding in Na is only reported in one EPMA image, but not conclusive in other maps (e.g. Fig. 1, Supplement section 2, 31-4). Potentially the expression of banding is also related to the absolute Na concentrations of the shell, as banding in *Ammonia tepida* (lower in Na) was less pronounced than in *Amphistegina*
*lessonnii* (Geerken et al., 2018). Accordingly the banding in planktonic foraminifera may also be less pronounced and hence not detectable within our approach. Irrespective, the relative contribution of these bands can be considered minor in comparison with the other zones of high Na/Ca values such as the spines and spine bases, which are clear also within the limited resolution of our analytical approach.

### 4.2 Unravelling spines and spine base Na/Ca

Several studies showed that Na/Ca in foraminiferal shells is not homogeneously distributed but is present at higher concentrations in bands and also at the (bases of) spine(s) (Branson et al., 2016;Mezger et al., 2018). Accordingly, Na-hotspots at spines and spine bases were selected to quantify Na/Ca values and compare these values with Na/Ca measured on shell-only areas. Furthermore, the preservation state of spine bases were studied, as these might still partially remain after spine shedding processes (Bé, 1980).

Spines sticking out of the shell showed Na/Ca to be consistently much higher than shell Na/Ca values from the same specimens. Spine base regions were selected based on backscattered and secondary electron images (Supplement section 1-3). Analyses from spine base areas, however, are likely influenced by mixing with lower Na/Ca values from adjacent regions. During EPMA analyses, the electron beam excites both areas/volumes in the region close to the interface between spine base and surrounding low-Na shell calcite. Moreover, EPMA analyses target a 2-D surface, whereas the spine is not necessarily oriented
parallel to the sampling surface. Hence, also in three dimensions variable amounts of spine-base related carbonate is analyzed during EPMA. Furthermore, due to its conic shape, spine thickness decreases towards the edges of the spine, also within the

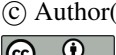


spine base. The sampling volume and pixel size together determines the obtained spine and spine-base Na/Ca signal. Therefore, the center of the spine - being the thickest and probably least affected by the polishing process - most likely reflects true spine base Na/Ca values. To estimate these signals, a mathematical approach was used in which we narrow the area perpendicular to

the center of the spine base for determining the Na/Ca (Fig. 6, Fig. S3). Narrowing the width of the spine base sampling area results in increasing Na/Ca values until they approach a plateau, which is assumed to reflect the true spine base Na/Ca value (Fig. 6a,b, Fig. S3). When no plateau is observed, e.g. the analyzed cross section is too small, true Na/Ca may remain unknown (Fig. 6a). Conversely, when increasing the width of the region used for calculating average values, values converge towards the shell values signal (Fig. 6a,b). As a result of decreasing the area used for calculating the average Na/Ca, standard errors increase

(Fig. 6a,b).

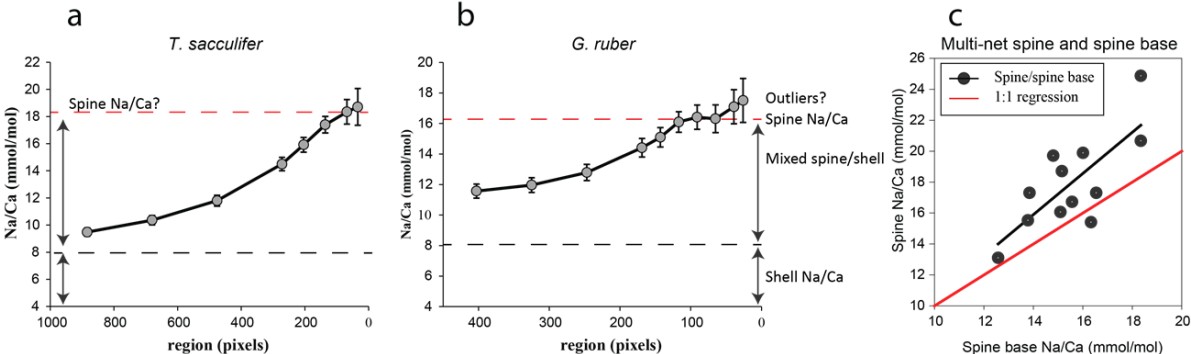

Figure 6: Examples of results of our spine-base quantification method, with: a): example of the quantification of a spine base (*T. sacculifer*, 2_12_multinet, Table 3), b) similar for *G. ruber*, (4_13_multinet, Table 4). The smaller/narrower the selected region, the higher the Na/Ca values, resulting from confining the analysis to the core of the spine. Due to the decreasing sampling volume,
standard errors also increase. Panel c) shows the spine Na/Ca values measured, versus the spine base Na/Ca obtained with the method described in section 6, Figure S3 and discussion section 4.2.

Based on our quantification approach, spine base Na/Ca values for the water column collected specimens range from 12.6 to 18.4 mmol/mol for *T. sacculifer* and from 15.0 to16.1 mmol/mol for *G. ruber* (Table 6). Compared to Na/Ca values of
the spines from the same specimens, spine base values are 4 to 35% lower (Fig. 6c). Although this offset is consistent and considerable, it cannot be excluded that it is primarily caused by the fundamental complication of estimating Na/Ca values in the spine bases. Whereas the spine bases are surrounded by low Na/Ca carbonate, spines are surrounded by the embedding material only, which does not affect the analyses. For the spine bases of specimens of the Red Sea surface water, sediment surface and cultured *T. sacculifer* specimens, Na/Ca values vary from 9.6 to 20 mmol/mol, with averages being consistently lower compared
to laser ablation measured spines (Mezger et al., 2018) and parts of the spines still sticking out after embedding, measured here with EPMA (Fig. 6c, Table 6).

The relatively high concentration of Na in spines and spine bases may be attributed to relatively fast growth rates compared to shell carbonate. Inorganic precipitation experiments suggest that growth rate enhances incorporation of most minor and trace metals, including Na (Busenberg and Plummer, 1985). Moreover, analogous to spine formation in sea urchins, an
amorphous precursor may be responsible for the prismatic shape of the foraminiferal spines, which rapidly transforms into calcite (Beniash et al., 1997). Such a precursor phase has also been shown by (Jacob et al., 2017) to occur during foraminiferal shell calcification, with formation of vaterite. They also suggest that an amorphous precursor may have been present in two planktonic foraminiferal species (Jacob et al., 2017). Such an amorphous calcium carbonate likely contains much more minor and trace elements, although a subsequent phase transformation from amorphous calcium carbonate (ACC) to calcite (potentially
via vaterite) would still affect element incorporation (Littlewood et al., 2017). Interestingly, this would not only influence Na





incorporation, but also most other minor and trace metals. Application of foraminiferal trace metals for proxy reconstructions should, therefore, also address the potential effect of differences in spine chemistry.

The consistently lower Na concentration of the spine base compared to the actual spines suggests a gradual transition from low-Na/Ca of the shell calcite to the high-Na/Ca of the spine (Fig. 6c, Fig. 7). Although our approach does not allow to

fully exclude an analytical bias, alternatively leakage or diffusion of Na from the high-Na spine base to the low-Na shell through time (Yoshimura et al., 2017) could also explain (part of) the observed intermediate values (Fig. 7). The spine would not be affected, or even has higher Na concentrations, as Na diffusion from seawater into the spine after or during spine formation could increase the Na content of the spine. This would increase the observed shell, spine-base to spine concentration gradient, but not influence the average whole-shell Na composition. Using synchrotron X-ray spectroscopy, (Yoshimura et al., 2017) found that

Na incorporation is associated with substitution for Ca in the calcite lattice. This is in contrast to what was proposed earlier by e.g. *Ishikawa and Ichikuni* (1984) and with the charge difference between $Na^+$ and $Ca^{2+}$ being compensated by the creation of $CO_3^{2-}$ vacancies (Yoshimura et al., 2017). These vacancies in the crystal lattice result in weaker calcite lattice spots at the locations of Na incorporation, facilitating leaching of Na from the calcite crystal on geological timescales (Yoshimura et al., 2017). However, in this study the Na/Ca composition of the foraminiferal shells (shell-only) of the same species (Red Sea water column and core-tops, as well as cultured specimens) were here found to remain similar (Fig. 5), implying no appreciable Na

exchange on these relatively short time scales (thousands of years). Still, it is not clear whether the spines, with relatively high Na concentrations and hence weak calcite lattice spots and a large surface to volume ratio, have been affected.



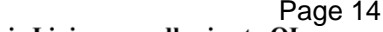

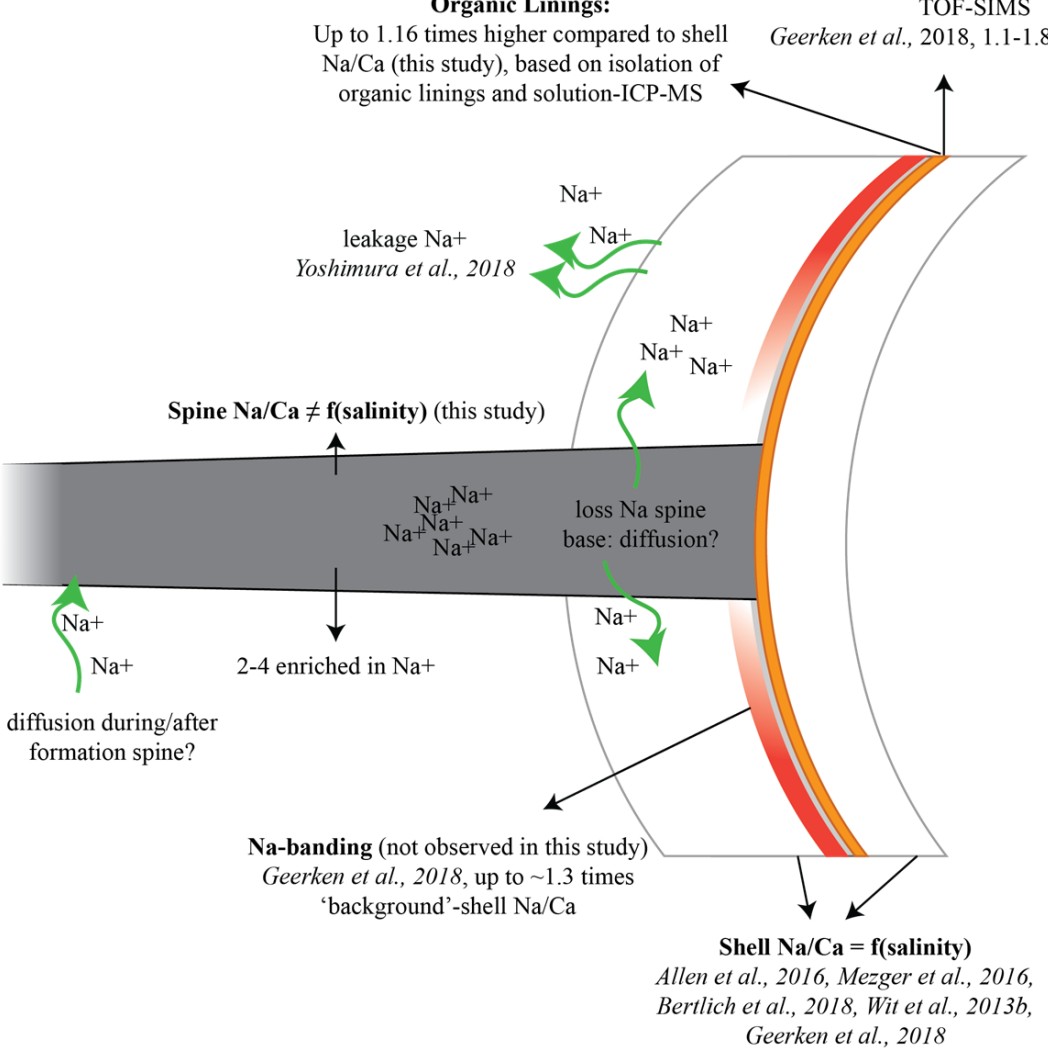

**Figure 7: Potential processes influencing the Na/Ca composition of the shell, spines and organic linings. Na from spines may be transported to the surrounding seawater or into the adjacent calcite with a relatively low Na/Ca. The latter may be difficult to distinguish from mixing of material from the spine and that of the low-Na/Ca calcite.**

## 4.3 Consequences of differences in spine and shell chemistry

The differences in Na/Ca between spine, spine base and shell-only carbonate can potentially explain differences observed between calibrations ((Mezger et al., 2018); Fig. 5). Differences between calibrations are observed in absolute Na/Ca concentrations and also between the slopes as a function of salinity. When spines are fully responsible for the observed difference in both slopes and absolute Na/Ca values between e.g. the cultured *T. sacculifer*, measured shell-only with EPMA



(Bertlich et al., 2018), and planktonic foraminifera with spines (Fig. 5, (Mezger et al., 2018)), this implies that either Na/Ca$_{spines}$ must increase with increasing salinities and/or the relative contribution of spine carbonate to the integrated whole test signal must increase with increasing salinity (Fig. 8). In case of the latter explanation this can be due to relative changes in spine-density, - thickness and/or -length compared to the thickness of the shell wall (Fig. 8).

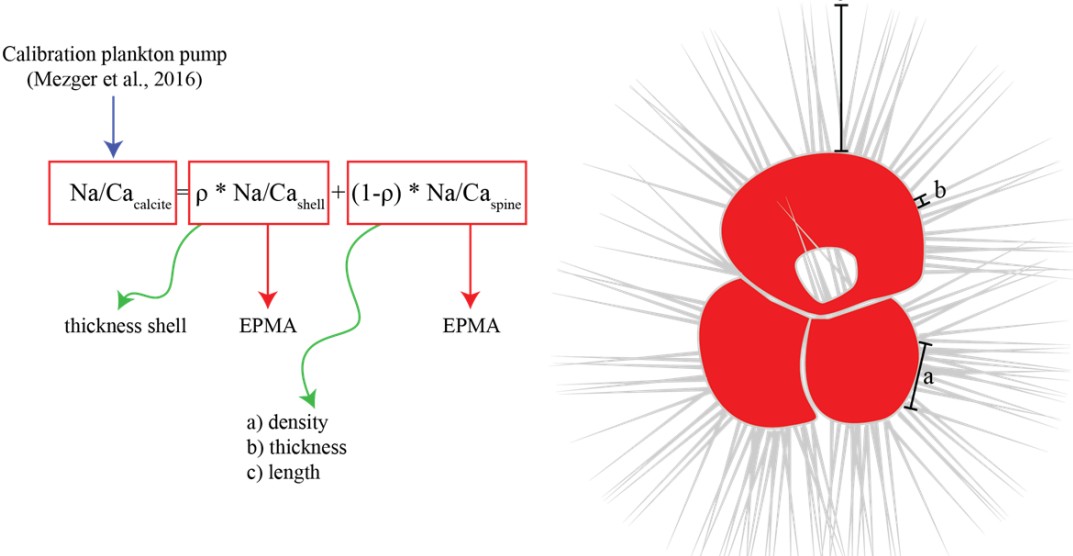

**Figure 8: General mass balance model combining relative contribution (ρ) of spine (Na/Ca$_{spine}$) and shell calcite (Na/Ca$_{shell}$) Na/Ca values to explain the whole shell (Na/Ca$_{calcite}$) Na/Ca values measured. The relative contribution of shell calcite depends on the shell thickness relative to the amount of spines, whereas the relative contribution of spines depends on spine density, thickness and length. The Na composition of the individual shell parts was measured with EPMA, and the total shell Na/Ca values are derived from laser ablation measurements on specimens still containing shell and spines (*Mezger et al.*, 2019). Spines could be up to 2-3 mm and are therefore not shown in their true scale in this image.**

Average Na/Ca calcite compositions of whole foraminiferal specimens reflect the relative contributions of Na/Ca in shell calcite (ρ) plus the contribution of Na/Ca in spine calcite (1-ρ) (Fig. 8). To determine the contribution of spines to the total Na/Ca$_{calcite}$ ('whole shell' Na/Ca) (Fig. 6), chamber-specific laser ablation-ICP-Q-MS Na/Ca values from Red Sea surface water collected *T. sacculifer* and *G. ruber* specimens (Mezger et al., 2016) are compared to the Na/Ca composition of shell-only EPMA-measured cultured *T. sacculifer* (Bertlich et al., 2018). Because the spines of surface dwelling foraminifera are still largely intact, the difference in absolute values and the slope between these calibrations allows calculating the relative contribution of spine bound Na to whole shell Na/Ca values (Fig. 5 and 8). To compare the exponential calibrations of *G. ruber* and *T. sacculifer*, the calibration of (Bertlich et al., 2018), was extrapolated with an exponential calibration. The relative contribution of spines to the total Na/Ca, based on LA-ICP-MS measured spine Na/Ca at a salinity of 39.6 (27.5 mmol/mol Na/Ca for *G. ruber* and 28.5 mmol/mol for *T. sacculifer*,(Mezger et al., 2018)) as well as EPMA-based spine Na/Ca at a salinity of 39.8 (on average 15.64 mmol/mol Na/Ca for *G. ruber* and 16.9 mmol/mol for *T. sacculifer*, was calculated based on the following equations:

$$\text{Na/Ca}_{\text{Mezger et al. 2016}} = \rho * \text{Na/Ca}_{\text{Bertlich et al. 2018}} + (1-\rho) * \text{Na/Ca}_{\text{spine}} \qquad (1)$$
$$\text{Na/Ca}_{\text{Mezger et al. 2016}} - \text{Na/Ca}_{\text{spine}} = \rho * (\text{Na/Ca}_{\text{Bertlich et al. 2018}} - \text{Na/Ca}_{\text{spine}}) \qquad (2)$$
$$\rho = (\text{Na/Ca}_{\text{Mezger et al. 2016}} - \text{Na/Ca}_{\text{spine}}) / (\text{Na/Ca}_{\text{Bertlich et al. 2018}} - \text{Na/Ca}_{\text{spine}}) \qquad (3)$$





This suggests a relative spine and spine base contribution from 20.8% (exp) to 19.75% (lin) for *G. ruber* and from 20.63% (exp) to 19.82% (lin) for *T. sacculifer*. However, when calculating the relative spine contribution from EPMA-based spine Na/Ca values, the relative spine contribution ranges from 46.7% (exp) to 43.3% (lin) for *G. ruber* and from 42.83% (exp) to 42.93% (lin) for *T. sacculifer*, which seem unrealistically high. To calculate the Na/Ca$_{spine}$ based on a constant ρ for different salinities (Fig. 8), the following equation is used:


$$(Na/Ca_{Mezger\ et\ al.\ 2016} - (ρ * Na/Ca_{Bertlich\ et\ al.\ 2018}))/ (1-ρ) = Na/Ca_{spine} \qquad (4)$$

Based on these calculations, LA-based spine Na/Ca values should increase 1.4 to 2.1 times (lin-exp *G. ruber*), and for *T. sacculifer* 1.4 to 2.2 times (lin-exp) within a natural salinity range from 30 to 40 to account for the difference in absolute values

between studies (Fig. 9). Alternatively, the concentration of spines (1-ρ) changes with increasing salinities (equation 3), from 8.6 to 21.6% or 19.9 to 27.8% for *G. ruber* (exp-lin) and from 7.9 to 21.4% or 13.4 to 20.1% for *T. sacculifer* (exp-lin).

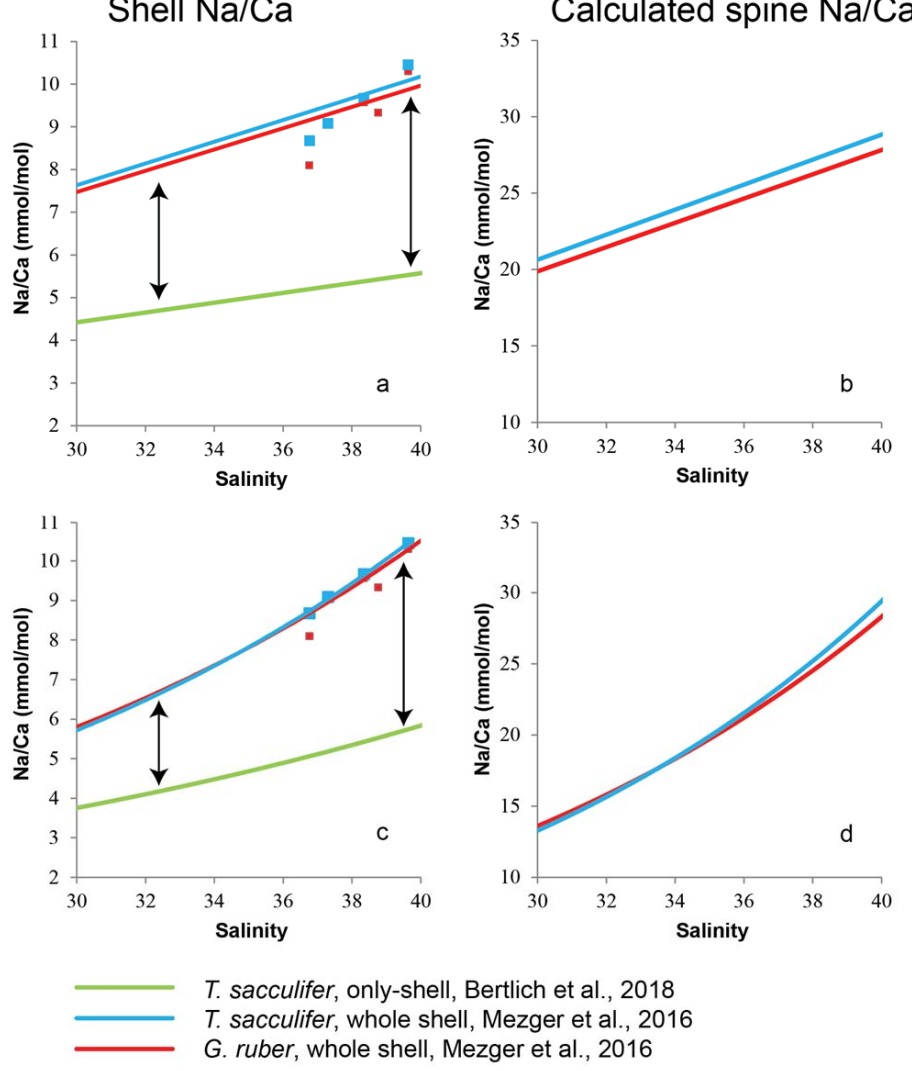

**Figure 9: Comparison of Na/Ca-salinity calibrations between surface water collected spinose planktonic specimens still containing spines (whole shell) (Mezger et al., 2016) and extrapolated cultured only-shell *T. sacculifer* (Bertlich et al., 2018). The difference in**



**absolute values is assumed to be caused by spines (arrow) (a). (b) Theoretically calculated Na/Ca values of the spines, in case of (a). (c)**
       **Difference in absolute values by exponential calibrations. (d) Theoretical calculated Na/Ca values of the spines, in case of (c).**

No appreciable change in number of spines (e.g. spine density) with increasing salinity has been observed and also the
width of the spines appears to decrease rather than increase with increasing salinity (Fig. 4). Spine length could vary with
salinity, but we were unable to quantify spine length as spines easily break off during sampling and sample processing. Spines

are connected to the planktonic foraminiferal shell through a thin organic lining, which is easily removed during cleaning. The
slight offset in absolute values between the cultured *T. sacculifer* and core-tops can be explained by spine bases, still partially
present in the shell wall after gametogenesis or burial.

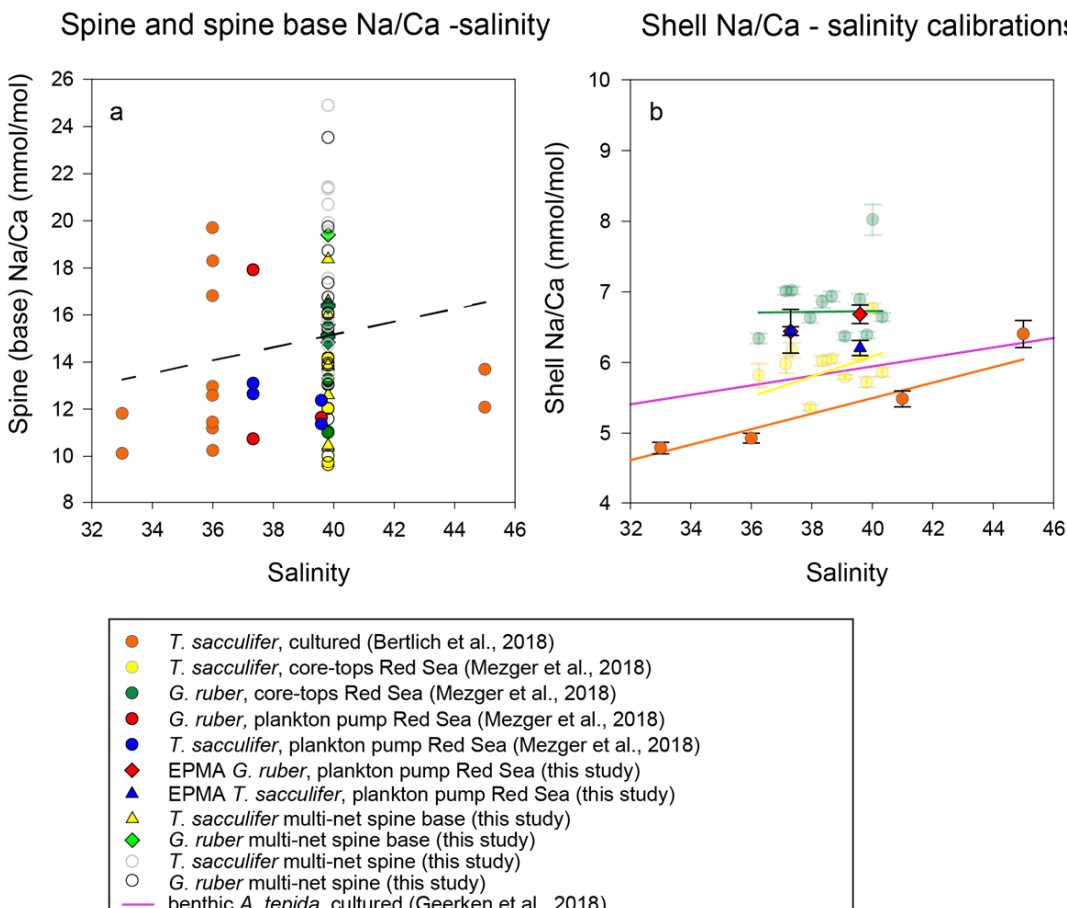

**Figure 10: (a) Changes of Na/Ca spine bases and spines (multi-net only) with salinity and (b) comparison of different existing Na/Ca**
**shell – salinity calibrations with indicated standard errors.**

Combining all spine and spine base Na/Ca values compared to ambient salinity, would suggest a trend towards higher
Na incorporation with higher salinities which is, however, not significant. Compared to shell Na/Ca composition of the same
specimens, spine Na/Ca values are 2-4 times higher. For Red Sea core-tops, no spines are observed (Mezger et al., 2018), and
SEM images often show spine holes, probably associated with life-stage related (gametogenesis) spine loss. Some spine bases

remain present, allowing quantification of core-top spine base Na/Ca. Comparing the EPMA measured spine and spine base
Na/Ca values with values calculated using a mass balance (see above, Fig.s 8-10) shows that measured absolute Na/Ca values are
lower and not in line with the calculated difference in slopes. This suggests that either 1) spine base Na/Ca does not vary with

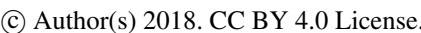


salinity, 2) EPMA measured values for spine-base and spine Na/Ca underestimate true spine values (Fig. 10) or 3) spine base Na/Ca values are significantly lower compared to the actual spine values. Although we here show a major impact of spines and

spine bases on Na/Ca, the Na/Ca values of the shell itself seem relatively robust (Fig. 10). Comparing both shell and spine Na/Ca values with salinity shows that shell chemistry records salinity, albeit with a very modest slope (Fig. 5, 10).

## Conclusions

Whole shell Na/Ca values, including spines and spine bases, show an offset to shell-only values due to the (variable) contribution of spine and spine base related carbonate, enriched in Na. Both absolute values and its relation to salinity show an offset between

specimens with and without spines. Whereas the high Na areas may be susceptible to taphonomic or ontogenetic alteration, the chemistry of the shell itself appears relatively robust. The Na composition of foraminiferal organic linings is, although higher than shell Na/Ca, not sufficient to significantly influence the overall Na/Ca values measured. Spine Na/Ca values, nor their width or density appears to respond to changes in salinity. However, potential effects of diffusion or sampling volume errors related to EPMA could also have resulted in somewhat lower spine base compared to spine Na/Ca values. Comparing both shell and spine

Na/Ca values with salinity shows that shell-only values still record salinity, albeit with a low sensitivity. This is relevant for the paleo-application of Na/Ca in reconstructing salinity since spines may not always preserve well.

*Data availability* All data on which this publication is based can be found in the tables in the manuscript and through the following doi: https://doi.org/10.4121/uuid:4aca8e7d-7e42-448b-9a77-f62c61e85049.

*Author contributions* GJR, LdN and EMM designed this study. All data preparations, measurements and interpretations were
executed by EMM, with daily discussions with GJR and LdN. Part of the measurements (culture experiments) was measured together with JBe. These culture experiments were carried out by JBi. EMM analyzed the data and prepared the manuscript, with contributions from all co-authors.

*Competing interests* The authors declare that they have no competing interests.

*Acknowledgements* No acknowledgements during submittal phase.




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



**Table 1**. List of characteristics of the EPMA measurements, excluding the multi-nets. The phrase 'in-situ' indicates that the measured chambers were not grown in culture, but formed in the natural environment 1-2 miles off the south coast of Curaçao (S=~35.9) before sampling. 'PP' refers to Red Sea plankton pump samples, 'CT' to Red Sea core-tops samples and 'exp' to experiments.

| Position | Species | PP/CT/exp | Magnification | HFW (total picture pixel width in μm) | pixel size (μm) | T (°C) | S |
|---|---|---|---|---|---|---|---|
| 31Jan_1 | *G. ruber* | PP | 2500 | 48 | 0.24 | 29.4 | 37.3 |
| 31Jan_2 | *G. ruber* | PP | 2500 | 48 | 0.24 | 29.4 | 37.3 |
| 31Jan_3 | *T. sacculifer* | PP | 2500 | 48 | 0.24 | 29.4 | 37.3 |
| 31Jan_4 | *T. sacculifer* | PP | 2500 | 48 | 0.27 | 29.4 | 37.3 |
| 31Jan_5 | *T. sacculifer* | PP | 2500 | 48 | 0.24 | 29.4 | 37.3 |
| 31Jan_6 | *T. sacculifer* | PP | 2500 | 48 | 0.27 | 26.3 | 39.6 |
| 31Jan_7 | *G. ruber* | CT | 2500 | 48 | 0.24 | 26.7 | 39.8 |
| 31Jan_8 | *T. sacculifer* | CT | 1600 | 75 | 0.38 | 26.7 | 39.8 |
| 31Jan_9 | *T. sacculifer* | CT | 2300 | 52 | 0.26 | 26.7 | 39.8 |
| 2Feb_1 | *G. ruber* | CT | 1700 | 48 | 0.24 | 26.7 | 39.8 |
| 2Feb_2 | *G. ruber* | PP | 2000 | 54 | 0.27 | 26.3 | 39.6 |
| 1Feb_1 | *T. sacculifer* | T-exp | 1400 | 86 | 0.43 | 23.5 | 36 |
| 1Feb_2 | *T. sacculifer* | T-exp | 2000 | 60 | 0.30 | 19.5 | 33 |
| 1Feb_3 | *T. sacculifer* | T-exp | 2500 | 48 | 0.24 | 19.5 | 33 |
| 1Feb_4 | *T. sacculifer* | T-exp | 1700 | 71 | 0.35 | in-situ | in-situ |
| 1Feb_5 | *T. sacculifer* | T-exp | 1400 | 86 | 0.34 | 26.5 | 33 |
| 2Feb_1 | *T. sacculifer* | S-exp | 1700 | 71 | 0.35 | in-situ | in-situ |
| 2Feb_2 | *T. sacculifer* | S-exp | 2222 | 54 | 0.30 | in-situ | in-situ |
| 2Feb_3 | *T. sacculifer* | S-exp | 2500 | 48 | 0.24 | in-situ | in-situ |
| 2Feb_4 | *T. sacculifer* | S-exp | 2222 | 54 | 0.30 | 26.5 | 45 |
| 2Feb_5 | *T. sacculifer* | S-exp | 1800 | 67 | 0.33 | in-situ | in-situ |
| 2Feb_6 | *T. sacculifer* | S-exp | 2200 | 55 | 0.27 | in-situ | in-situ |



**Table 2.** Overview EPMA shell measurements of different parts of the shell for core-tops and plankton pumps.

| Species | PP/CT | Salinity | #Pixels | Na/Ca mean (mmol/mol) | Na/Ca median (mmol/mol) | SD | SE |
|---|---|---|---|---|---|---|---|
| *G. ruber* | CT | 39.8 | 1512 | 5.45 | 4.28 | 3.55 | 0.10 |
| *G. ruber* | CT | 39.8 | 810 | 5.45 | 5.32 | 3.42 | 0.13 |
| *G. ruber* | CT | 39.8 | 549 | 5.41 | 4.05 | 3.75 | 0.17 |
| *G. ruber* | CT | 39.8 | 630 | 5.76 | 4.48 | 3.70 | 0.16 |
| *T. sacculifer* | CT | 39.8 | 310 | 6.02 | 5.63 | 3.91 | 0.23 |
| *T. sacculifer* | CT | 39.8 | 368 | 6.18 | 5.83 | 3.97 | 0.21 |
| *T. sacculifer* | CT | 39.8 | 180 | 5.69 | 4.10 | 4.14 | 0.31 |
| *T. sacculifer* | CT | 39.8 | 405 | 6.22 | 5.91 | 3.98 | 0.23 |
| *T. sacculifer* | CT | 39.8 | 288 | 5.98 | 5.90 | 3.76 | 0.24 |
| *T. sacculifer* | CT | 39.8 | 864 | 5.52 | 4.25 | 3.74 | 0.14 |
| *T. sacculifer* | CT | 39.8 | 851 | 5.78 | 4.37 | 3.89 | 0.14 |
| *G. ruber* | CT | 39.8 | 288 | 6.84 | 6.34 | 4.16 | 0.25 |
| *G. ruber* | CT | 39.8 | 350 | 6.72 | 6.27 | 4.22 | 0.23 |
| *G. ruber* | CT | 39.8 | 774 | 6.36 | 5.96 | 3.95 | 0.15 |
| *G. ruber* | CT | 39.8 | 644 | 6.39 | 6.00 | 4.03 | 0.17 |
| *G. ruber* | PP2 | 37.3 | 420 | 5.75 | 5.40 | 3.78 | 0.20 |
| *G. ruber* | PP2 | 37.3 | 444 | 5.66 | 4.27 | 3.84 | 0.20 |
| *G. ruber* | PP2 | 37.3 | 468 | 5.89 | 5.40 | 4.03 | 0.20 |
| *G. ruber* | PP2 | 37.3 | 468 | 5.60 | 4.31 | 3.72 | 0.18 |
| *G. ruber* | PP2 | 37.3 | 546 | 6.62 | 5.91 | 4.38 | 0.20 |
| *G. ruber* | PP2 | 37.3 | 420 | 6.61 | 6.02 | 4.12 | 0.20 |
| *G. ruber* | PP2 | 37.3 | 476 | 7.73 | 6.98 | 5.03 | 0.25 |
| *G. ruber* | PP2 | 37.3 | 476 | 7.66 | 7.27 | 4.98 | 0.25 |
| *T. sacculifer* | PP2 | 37.3 | 406 | 6.77 | 6.20 | 4.50 | 0.14 |
| *T. sacculifer* | PP2 | 37.3 | 338 | 6.81 | 6.34 | 4.22 | 0.17 |
| *T. sacculifer* | PP2 | 37.3 | 351 | 6.29 | 5.97 | 4.03 | 0.15 |
| *T. sacculifer* | PP2 | 37.3 | 450 | 6.86 | 6.18 | 4.26 | 0.17 |
| *T. sacculifer* | PP2 | 37.3 | 540 | 6.12 | 5.75 | 3.90 | 0.20 |
| *T. sacculifer* | PP2 | 37.3 | 1584 | 6.37 | 5.97 | 4.09 | 0.23 |
| *T. sacculifer* | PP2 | 37.3 | 858 | 6.32 | 5.93 | 3.82 | 0.22 |
| *T. sacculifer* | PP2 | 37.3 | 720 | 6.50 | 5.99 | 4.41 | 0.22 |
| *T. sacculifer* | PP2 | 37.3 | 768 | 6.36 | 5.95 | 4.01 | 0.18 |
| *T. sacculifer* | PP2 | 37.3 | 858 | 6.83 | 6.10 | 4.48 | 0.13 |
| *T. sacculifer* | PP2 | 37.3 | 756 | 6.54 | 5.98 | 4.23 | 0.15 |
| *T. sacculifer* | PP2 | 37.3 | 204 | 6.48 | 5.94 | 4.21 | 0.16 |
| *T. sacculifer* | PP7 | 39.6 | 357 | 6.44 | 5.72 | 4.44 | 0.17 |
| *T. sacculifer* | PP7 | 39.6 | 476 | 6.49 | 5.97 | 4.41 | 0.15 |
| *T. sacculifer* | PP7 | 39.6 | 261 | 6.12 | 5.52 | 4.01 | 0.15 |
| *T. sacculifer* | PP7 | 39.6 | 208 | 6.75 | 6.25 | 4.51 | 0.32 |
| *G. ruber* | PP7 | 39.6 | 1053 | 6.20 | 5.88 | 4.08 | 0.23 |
| *G. ruber* | PP7 | 39.6 | 735 | 5.91 | 5.60 | 3.88 | 0.20 |
| *G. ruber* | PP7 | 39.6 | 780 | 6.32 | 5.93 | 3.99 | 0.26 |
| *G. ruber* | PP7 | 39.6 | 6952 | 6.39 | 6.05 | 4.07 | 0.29 |





**Table 3.** Overview of multi-net Na/Ca shell values. Similar colors indicate that these are measurements from the same specimen (except for the white color).

| Position | Species | #Pixels | Na/Ca mean (mmol/mol) | Na/Ca median (mmol/mol) | SD | SE |
|---|---|---|---|---|---|---|
| 0001_1 | *T. sacculifer* | 2898 | 7.09 | 6.77 | 4.58 | 0.10 |
| 0001_1 | *T. sacculifer* | 1568 | 6.96 | 6.33 | 4.78 | 0.13 |
| 0002_1 | *T. sacculifer* | 903 | 6.61 | 5.67 | 4.32 | 0.15 |
| 0002_3 | *T. sacculifer* | 1060 | 6.76 | 6.69 | 4.14 | 0.13 |
| 0002_4 | *T. sacculifer* | 2652 | 6.13 | 5.27 | 4.06 | 0.08 |
| 0002_8 | *T. sacculifer* | 609 | 6.08 | 5.17 | 4.25 | 0.17 |
| 0002_8 | *T. sacculifer* | 587 | 6.30 | 5.25 | 4.79 | 0.20 |
| 0002_11 | *T. sacculifer* | 1109 | 6.79 | 6.54 | 4.38 | 0.13 |
| 0002_12 | *T. sacculifer* | 567 | 5.80 | 5.09 | 4.18 | 0.18 |
| 0002_12 | *T. sacculifer* | 1567 | 5.73 | 4.89 | 4.10 | 0.10 |
| 0002_13 | *T. sacculifer* | 1150 | 5.55 | 4.91 | 3.93 | 0.12 |
| 0004_24 | *T. sacculifer* | 966 | 6.26 | 5.50 | 4.18 | 0.13 |
| 0004_7 | *G. ruber* | 777 | 8.42 | 7.43 | 5.04 | 0.18 |
| 0004_7 | *G. ruber* | 913 | 8.07 | 7.52 | 4.84 | 0.16 |
| 0004_12 | *G. ruber* | 1299 | 6.82 | 6.65 | 4.32 | 0.12 |
| 0004_13 | *G. ruber* | 1195 | 6.90 | 6.65 | 4.37 | 0.13 |
| 0004_14 | *G. ruber* | 1361 | 7.12 | 6.81 | 4.27 | 0.12 |
| 0004_15 | *G. ruber* | 1013 | 5.95 | 5.18 | 4.08 | 0.13 |
| 0004_16 | *G. ruber* | 2479 | 6.78 | 6.58 | 4.47 | 0.09 |
| 0004_17 | *G. ruber* | 1248 | 6.88 | 6.47 | 4.62 | 0.13 |
| 0004_18 | *G. ruber* | 1665 | 5.97 | 5.04 | 4.09 | 0.10 |
| 0004_19 | *G. ruber* | 439 | 6.29 | 5.33 | 4.13 | 0.20 |
| 0004_22 | *G. ruber* | 850 | 6.64 | 6.23 | 4.55 | 0.16 |






**Table 4.** Overview of spine and spine base Na/Ca values *T. sacculifer* multi-nets. Similar colors indicate that these are measurements from the same specimen.

| Position | Species | spine/base | #Pixels | Na/Ca mean (mmol/mol | Na/Ca median (mmol/mol) | SD | SE |
|---|---|---|---|---|---|---|---|
| 0001_1 | *T. sacculifer* | base | 160 | 12.57 | 11.70 | 6.15 | 0.49 |
| 0002_1 | *T. sacculifer* | spine | 621 | 13.10 | 12.72 | 6.06 | 0.24 |
| 0002_3 | *T. sacculifer* | base | 69 | 13.78 | 13.97 | 6.48 | 0.78 |
| 0002_3 | *T. sacculifer* | spine | 220 | 15.52 | 15.42 | 7.06 | 0.67 |
| 0002_4 | *T. sacculifer* | spine 1 | 91 | 17.30 | 17.15 | 6.79 | 0.73 |
| 0002_4 | *T. sacculifer* | base 1 | 98 | 13.84 | 13.04 | 6.16 | 0.63 |
| 0002_4 | *T. sacculifer* | base 2 | 50 | 16.54 | 16.78 | 6.53 | 0.92 |
| 0002_5 | *T. sacculifer* | spine | 345 | 19.89 | 19.43 | 7.59 | 0.41 |
| 0002_5 | *T. sacculifer* | base | 64 | 16.01 | 15.89 | 6.75 | 1.05 |
| 0002_6 | *T. sacculifer* | spine | 234 | 19.53 | 18.74 | 7.47 | 0.49 |
| 0002_7 | *T. sacculifer* | spine | 97 | 21.41 | 20.27 | 8.23 | 0.84 |
| 0002_8 | *T. sacculifer* | spine 1 | 98 | 21.33 | 21.35 | 8.29 | 0.84 |
| 0002_8 | *T. sacculifer* | spine 2 | 141 | 15.97 | 15.44 | 7.94 | 0.67 |
| 0002_9 | *T. sacculifer* | spine | 190 | 15.69 | 14.73 | 7.75 | 0.56 |
| 0002_10 | *T. sacculifer* | spine | 209 | 14.68 | 14.08 | 6.75 | 0.47 |
| 0002_11 | *T. sacculifer* | spine | 502 | 17.53 | 17.24 | 7.09 | 0.32 |
| 0002_12 | *T. sacculifer* | spine 1 | 43 | 20.67 | 23.73 | 8.43 | 1.29 |
| 0002_12 | *T. sacculifer* | base 1 | 189 | 10.35 | 9.24 | 5.67 | 0.42 |
| 0002_12 | *T. sacculifer* | spine 2 | 25 | 24.87 | 25.90 | 9.71 | 1.94 |
| 0002_12 | *T. sacculifer* | base 2 | 68 | 18.35 | 18.14 | 7.16 | 0.91 |
| 0002_13 | *T. sacculifer* | base | 70 | 10.43 | 8.98 | 5.99 | 0.72 |
| 0002_13 | *T. sacculifer* | spine | 1230 | 21.40 | 20.93 | 8.35 | 0.24 |
| 0004_24 | *T. sacculifer* | spine | 362 | 13.45 | 13.10 | 6.73 | 0.35 |





**Table 5.** Overview of spine and spine base Na/Ca values *G. ruber* multi-nets. Similar colors indicate that these are measurements from the same specimen (except for white, these are different specimens).

| Position | Species | spine/base | #Pixels | Na/Ca mean (mmol/mol | Na/Ca median (mmol/mol) | SD | SE |
|---|---|---|---|---|---|---|---|
| 0002_14 | *G. ruber* | spine | 47 | 17.34 | 15.86 | 1.18 | 8.11 |
| 0004_1 | *G. ruber* | spine | 28 | 9.98 | 8.60 | 6.95 | 1.31 |
| 0004_4 | *G. ruber* | spine | 55 | 13.83 | 13.24 | 6.80 | 0.91 |
| 0004_5 | *G. ruber* | spine | 41 | 14.12 | 13.60 | 7.27 | 1.14 |
| 0004_6 | *G. ruber* | spine | 36 | 13.03 | 11.94 | 5.41 | 0.90 |
| 0004_7 | *G. ruber* | base | 42 | 19.38 | 18.02 | 5.72 | 0.88 |
| 0004_10 | *G. ruber* | spine | 94 | 11.55 | 11.30 | 6.56 | 0.68 |
| 0004_11 | *G. ruber* | spine | 53 | 23.50 | 20.80 | 14.02 | 1.92 |
| 0004_12 | *G. ruber* | base | 150 | 15.15 | 14.44 | 6.41 | 0.53 |
| 0004_12 | *G. ruber* | spine | 31 | 18.70 | 18.37 | 6.63 | 1.19 |
| 0004_13 | *G. ruber* | spine | 30 | 19.71 | 18.98 | 7.47 | 1.36 |
| 0004_13 | *G. ruber* | base | 91 | 16.41 | 15.97 | 7.12 | 0.80 |
| 0004_14 | *G. ruber* | base | 100 | 15.01 | 13.70 | 7.33 | 0.73 |
| 0004_15 | *G. ruber* | base | 124 | 14.81 | 14.60 | 6.49 | 0.59 |
| 0004_16 | *G. ruber* | spine | 53 | 16.72 | 16.42 | 5.26 | 0.72 |
| 0004_16 | *G. ruber* | base | 108 | 15.57 | 14.82 | 7.04 | 0.73 |
| 0004_17 | *G. ruber* | spine | 64 | 15.41 | 15.09 | 6.92 | 0.87 |
| 0004_17 | *G. ruber* | base | 108 | 16.33 | 15.48 | 6.36 | 0.66 |
| 0004_18 | *G. ruber* | spine | 41 | 16.06 | 14.38 | 8.16 | 1.27 |
| 0004_19 | *G. ruber* | base | 128 | 15.10 | 13.66 | 8.13 | 0.76 |
| 0004_20 | *G. ruber* | spine | 53 | 12.01 | 11.53 | 5.90 | 0.81 |
| 0004_21 | *G. ruber* | spine | 21 | 16.01 | 15.45 | 7.28 | 1.59 |
| 0004_22 | *G. ruber* | spine | 75 | 13.92 | 12.72 | 6.13 | 0.71 |




**Table 6.** Overview spine base measurements of cultured and field-collected specimens (Table 1)

| Position | Species | Salinity | #Pixels | Na/Ca mean (mmol/mol | Na/Ca median (mmol/mol) | SD | SE |
|---|---|---|---|---|---|---|---|
| 1Feb_1 | *T. sacculifer* | 36.0 | 59 | 18.29 | 17.05 | 6.97 | 0.91 |
| 1Feb_1 | *T. sacculifer* | 36.0 | 90 | 10.24 | 9.14 | 6.06 | 0.67 |
| 1Feb_3 | *T. sacculifer* | 33.0 | 50 | 11.82 | 11.35 | 5.56 | 0.79 |
| 1Feb_4 | *T. sacculifer* | 36.0 | 160 | 19.70 | 18.90 | 8.64 | 0.69 |
| 1Feb_5 | *T. sacculifer* | 33.0 | 174 | 10.12 | 10.29 | 5.31 | 0.41 |
| 2Feb_2 | *T. sacculifer* | 36.0 | 98 | 16.81 | 16.13 | 8.26 | 0.84 |
| 2Feb_3 | *T. sacculifer* | 36.0 | 70 | 12.96 | 12.80 | 6.77 | 0.81 |
| 2Feb_4 | *T. sacculifer* | 45.0 | 84 | 12.07 | 11.11 | 6.63 | 0.75 |
| 2Feb_4_spine2 | *T. sacculifer* | 45.0 | 56 | 13.69 | 13.71 | 6.84 | 0.94 |
| 2Feb_5 | *T. sacculifer* | 36.0 | 150 | 12.58 | 10.56 | 9.36 | 0.98 |
| 2Feb_6_spine1 | *T. sacculifer* | 36.0 | 70 | 11.19 | 9.68 | 6.41 | 0.80 |
| 2Feb_6_spine2 | *T. sacculifer* | 36.0 | 117 | 11.44 | 9.88 | 6.10 | 0.79 |
| 31Jan_1 | *G. ruber* | 37.3 | 153 | 10.70 | 10.72 | 6.11 | 0.91 |
| 31Jan_2 | *G. ruber* | 37.3 | 24 | 17.90 | 17.36 | 6.32 | 1.29 |
| 31Jan_3 | *T. sacculifer* | 37.3 | 48 | 12.64 | 10.92 | 6.51 | 0.94 |
| 31Jan_4 | *T. sacculifer* | 37.3 | 50 | 13.10 | 11.21 | 6.86 | 0.97 |
| 31Jan_6 | *T. sacculifer* | 39.6 | 50 | 11.44 | 10.96 | 6.28 | 1.11 |
| 31Jan_6 | *T. sacculifer* | 39.6 | 72 | 12.36 | 12.65 | 5.97 | 0.71 |
| 3Feb_2 | *G. ruber* | 39.6 | 42 | 11.64 | 11.30 | 5.32 | 0.83 |
| 31Jan_7 | *G. ruber* | 39.8 | 45 | 13.01 | 13.43 | 7.38 | 1.11 |
| 31Jan_7 | *G. ruber* | 39.8 | 25 | 13.19 | 14.41 | 7.31 | 1.46 |
| 31Jan_7 | *G. ruber* | 39.8 | 36 | 13.24 | 11.68 | 6.92 | 1.17 |
| 31Jan_8 | *T. sacculifer* | 39.8 | 20 | 14.15 | 14.63 | 3.81 | 0.85 |
| 31Jan_8 | *T. sacculifer* | 39.8 | 30 | 11.01 | 8.77 | 6.69 | 1.26 |
| 31Jan_8 | *T. sacculifer* | 39.8 | 36 | 9.57 | 7.63 | 6.76 | 1.14 |
| 31Jan_8 | *T. sacculifer* | 39.8 | 36 | 9.71 | 8.12 | 6.00 | 1.04 |
| 31Jan_9 | *T. sacculifer* | 39.8 | 70 | 15.34 | 13.99 | 8.79 | 1.08 |
| 31Jan_9 | *T. sacculifer* | 39.8 | 44 | 11.99 | 11.26 | 8.24 | 1.24 |
| 3Feb_1 | *G. ruber* | 39.8 | 15 | 10.98 | 11.73 | 4.05 | 1.05 |





**Table 7.** Elemental composition of organic linings calculated for shell weight and estimated elemental composition based on OL weight.

| | Na | Mg | Sr |
|---|---|---|---|
| *G. ruber* | | | |
| ppm OL (average + SE) | 1389±29 | 9325±34 | 84±0.19 |
| average test El/Ca (mmol/mol)* | 6.42 | 4.2 | 1.63 |
| **relative contribution OL Na to total shell Na** | | | |
| ppm | 5.61 | 37.68 | 0.34 |
| mmol/mol | 0.024 | 0.16 | 0.0004 |
| % | 0.38 | 3.69 | 0.02 |
| *T. sacculifer* | | | |
| ppm OL (average + SE) | 1703±11 | 9798±24 | 34±0.04 |
| average test El/Ca (mmol/mol)* | 6.38 | 4.1 | 1.6 |
| **relative contribution OL Na to total shell Na** | | | |
| ppm | 12.03 | 69.25 | 0.24 |
| mmol/mol | 0.052 | 0.28 | 0.0003 |
| % | 0.82 | 6.95 | 0.02 |

* average shell Sr/Ca and Mg/Ca based on *Mezger et al.*, 2016, shell Na/Ca based on *Mezger et al.*, 2018