# Peer review of "Planktonic foraminiferal spine versus shell carbonate Na incorporation in relation to salinity"

_Biogeosciences, 2018_

## Referee Comment (RC1) · Toyofuku (Referee) · 13 Dec 2018

General considerations

In this study, detailed sodium (Na) content and distribution of planktonic foraminiferal test. Na distribution of spine and spine base had been not well studied by previous studies. This point is greatly progressed by this study. The authors also suceed to show that presence or absence of spine / spine base enriched with Na can have a significant effect on bulk Na / Ca. Geochemical analysis of numerous shells and organic linings and model calculations are carried out by this study. I can certify this is a good research that has been extensively studied about spine and spine-based Na / Ca, and it is expected that readers of Biogeoscience will be interested with this topic.

[Figure]

Questions and comments

Have authors not analyzed chlorine by EPMA? Also, please be sure to show the how many times and how long time the samples were wash with water (L. 94). In considering Na / Ca, salt NaCl is the most popular and strong contaminant from seawater. It is necessary to know whether the distribution of Na is related to Cl or independent. Authors insist that Na is included in lattice with cited literature, but may Na that is not included in lattice exist, too. It is also necessary to make sure whether there is a change in Na concentration depending on the time and number of washing with water (L. 98) to consider the influence of NaCl. The washing process never change Na contents?

Why authors did not analyze using samples with different salinity condtions to examine the relationship between Na / Ca of spine and spine base and salinity? (Section 4.3). In Mezger et al. (2016), the first author analyzed planktonic foraminiferal specimens collected by plankton pump in Red sea with salinity gradient. Therefore, it seems possible to analyze Na / Ca of spine compared with salinity with these samples. However, in this study, as shown in Fig. 8-10, from the consideration based on the model calculation, it is concluded that Na / Ca of spine does not correlate with water temperature. The approach by model calculation is interesting and the conclusion is logical. Robust discussion can be constructed if there is support by measurement. In the future study, I think there is a possibility that samples with different salinity conditions can be measured. For that reason, it is not bad idea to leave room for discussion in the conclusion.

L. 101: Gentle setting for measurement of Na. Did you check reproducibility of Na measurement with this setting on standard materials? Further, this voltage seems bit weak (not impossible) for calcium detection. Authors will explain about the meaning of optimization of measurement setting of EPMA.

L. 132: SEM 3000 means "Miniscope TM 3000"? I can not find this type of SEM on Internet.

L. 201; I prefer more informative subtitle of paragraph. e.g. SEM measurement of spine morphology and densities.

Figure 5: Can you indicate the difference in species by color? e.g. T. sacculifer is blue-green colors, G. ruber is red-orange colors. Describe the meanings of shown lines in this caption. Further, could you indicate which data were measured by EPAM or LA-ICP-MS?

4.1 Organic Linings: Authors could show that organic lining were poor with Na. Why EPMA Na mapping never show OM as low concentration bands?

Figure 6: It is not appropriate to show "?" here. Describe possible explanation for "Spine Na/Ca" and "Outliers" in the plot Instead of indicating "?".

L. 331: This is important consideration because the measured results are variable at part by part by such partial measuement method as LA-ICP-MS, SIMS, EPMA and more. The authors will mention the importance about the choice of measurement portion on the test.

Figure 7: It is good useful compilation figure of Na/Ca understanding. Could you explain all indicated knowledge of these in the main text? It would be fit the paragraph started from L333 or around the sentence.

L. 355: Some figure and/or previous study should be referred after "function of salinity".

L. 357 "either Na/C a spines must increase with increasing salinities"

Figure 10:What are the lines in the fig. 10a? Can you indicate Na/Ca of spine and Na/Ca of spine base in separated plots?

---

## Short Comment (SC1) · 2 Jan 2019

It is a pleasure to read the considerate and well-written paper of Mezger and coauthors on "Planktonic foraminiferal spine versus shell carbonate Na incorporation in relation to salinity". In addition to the review of Takashi Toyofuku's, I only have two more comprehensive points to be considered by the authors, as well as some small points to be taken into consideration before publication of the paper.

It would be interesting and important to know, which of the morphotypes of T. sacculifer, and which morpho- and genotypes of G. ruber are analyzed here and are represented by the data. Since some of the different types have different ecological demands, the new Na/Ca possibly represent very specific ecological conditions, which in turn may

be reconstructed when using Na/Ca as a proxy of past conditions. The data plotted in Figure 4 may even show more structure when being plotted separately for different morphotypes? Also, "variability within one species" (line 256) may turn out to make sense for different morphotypes (also lines 258-259) ?!

In section 3.2, Scanning electron microscopy, the authors may want to consider that there are two different types of spines both in G. ruber and T. sacculifer, one being round and the other being triangular (see, e.g., Schiebel and Hemleben, 2017). The very different spine widths reported in lines 205 to 208 may result from the two different types of round (thinner) and triangular (thicker) spines, and may hence represent two groups of data in each of the species, and, even more importantly, Na/Ca may vary between the two types of spines.

Minor points: Line 22: better refer to carbonate, not calcite, because of other carbonate species like vaterite (Jacob et al., 2017); in the entire manuscript Line 23: better use "taphonomic alteration" than "taponomy" (also line 79) Line 56: bivalves have no spines (Zhao et al. 2017), and I cannot see any connection here Line 108: the "Whole shell" is called "test" in foraminifera; why do the authors avoid the term in the entire manuscript? Line 171: s-pecimen Line 179: ruber italic Lines 197-198: "in mixing signals between spine base and shell carbonate", possibly resulting from the resolution of measurements Line 208: "Salinity correlates negatively with spine width", may be turned around to keep the right order Figure 5: please indicate which data refer to entire test and which to shell (-only) measurments: One comma too many in the caption. Line 355: better: "When spines fully account for. . ." Line 367: better "up to 2-3 mm long ..." Line 374: (Figs. 5 and 8), check in the entire manuscript Line 378: . . . sacculifer); open parentheses Line 395: Na concentration?

---

## Author Comment (AC1) · 16 Jan 2019

Dear editor and dr. Takashi Toyofuku,

Thank you very much for the useful comments, additions and suggestions on our manuscript. We have changed and/or answered these comments step-by-step in the text below, explaining why we agree or respectfully disagree. Our answers are directly below the individual comments.

Also on behalf of the other authors,

Respectfully yours,

Eveline Mezger

[Figure]

Takashi Toyofuku: "Uploaded my comments are bit older revision with unexpected re-action of website. I hope authors use my comments indicated below. *** General considerations In this study, detailed sodium (Na) content and distribution of planktonic foraminiferal test. Na distribution of spine and spine base had been not well studied by previous studies. This point is greatly progressed by this study. The authors also suceed to show that presence or absence of spine / spine base enriched with Na can have a significant effect on bulk Na / Ca. Geochemical analysis of numerous shells and organic linings and model calculations are carried out by this study. I can certify this is a good research that has been extensively studied about spine and spine-based Na / Ca, and it is expected that readers of Biogeoscience will be interested with this topic."

Questions and comments

1) Have you not analyzed chlorine by EPMA? Also, please be sure to show the how many times and how long the samples were wash with water (L. 94). In considering Na / Ca, salt NaCl is the most popular and strong contaminant from seawater. It is nec-essary to know whether the distribution of Na is related to Cl or independent. Authors insist that Na is included in lattice with cited literature, but may Na that is not included in lattice exist, too. It is also necessary to make sure whether there is a change in Na concentration depending on the time and number of washing with water (L. 98) to consider the influence of NaCl. The washing process never change Na contents?

Thank you for this suggestion. Unfortunately, this method does not allow for chlorine (Cl) measurements since their concentrations in calcite are much lower ($\sim$40 times) than those of sodium (Na). This has now been added to the text at line 107. The fact that Cl is much lower is still a valuable addition to our study. This implies that Na in-corporation into the shell as (micro-)fluid inclusions (similar to what is suggested for Cl, Kitano et al., 1975), only provides a very small contribution to the total Na. Therefore, the effect of rinsing and potentially losing the fluid-included Na, does not significantly affect our results. Clearly, Na is – as far as resolution permits – homogeneously dis-tributed throughout shell calcite, and higher in spine (base) calcite, with no (visible)

[Figure]

contamination on the outside of the shells. We rinsed every polished sample three times to avoid contaminants on the calcite surface. As these (polishing powder) particles were very loosely attached to the exposed surface, contact time with the deionized water was kept very brief and is not expected to influence lattice-bound Na. We now added this information to the manuscript (line 98). So far, leaching of Na from the calcite lattice has only been observed on much longer (Myr) timescales (Yoshimura et al., 2017). Furthermore, after extensively testing the effect of the number of rinsing steps on the Na-composition in inorganic calcite powder (Mezger et al., in preparation for submission), this did not show any difference in calcite Na/Ca (Figure R1).

Caption Figure R1: Na/Ca values of inorganic calcite powder, measured with iCAP-Q-ICP-MS at the Royal NIOZ, plotted as a function of cleaning steps with supersaturated CaCO3 solution and de-ionized water (here referred to as milli-Q). The relatively high Na/Ca before rinsing indicates seawater present at the surface of the precipitated calcites.

2) Why authors did not analyze samples of different salinity conditions to examine the relationship between Na / Ca of spine and spine base and salinity? (Section 4.3). In Mezger et al. (2016), the first author analyzed planktonic foraminiferal specimens collected by plankton pump in Red sea with salinity gradient. Therefore, it seems possible to analyze Na / Ca of spine compared with salinity with these samples. However, in this study, as shown in Fig. 8-10, from the consideration based on the model calculation, it is concluded that Na / Ca of spine does not correlate with water temperature. The approach by model calculation is interesting and the conclusion is logical. Robust discussion can be constructed if there is support by measurement. In the future study, I think there is a possibility that samples with different salinity conditions can be measured. For that reason, it is not bad idea to leave room for discussion in the conclusion.

Thank you very much for this useful comment. Unfortunately, we did not have enough material left to study the chemical (EPMA-measured) shell composition for the whole

salinity range (Mezger et al., 2016). These surface water collected specimens were very thin and fragile, and therefore often severely damaged upon laser ablation analyses. Furthermore, due to cleaning procedures, different life stages and the vulnerability of spines, it was challenging to assemble enough material to measure spine composition at all. Spine compositions could only be measured reliably on 'new' multinet collected specimens, which were cleaned differently compared to the other samples used in previous studies. The relationship between salinity and shell Na was based so far on averages of multiple measurements on many different specimens, as the inter- and intra-specimen Na variability is quite large. In this study, we analyze a limited number of individuals and therefore, more specimens and measurements are probably necessary to investigate the potential relationship between spine Na and salinity. We added an extra sentence on this discussion subject in line 428.

3) L. 101: Gentle setting for measurement of Na. Did you check reproducibility of Na measurement with this setting on standard materials? Further, this voltage seems bit weak (not impossible) for calcium detection. Authors will explain about the meaning of optimization of measurement setting of EPMA.

The JCP-1 standard was measured multiple times (n=6) with the same (optimized) settings as the samples, we now clarified this at line 130. The gentle settings of the EPMA allowed us to increase the measurement time on the calcite, since the material and resin is too delicate for more intense beam settings.

4) L. 132: SEM 3000 means "Miniscope TM 3000"? I can not find this type of SEM on Internet.

You are correct, the official name would be 'Hitachi High-Tech TM3000 TableTop scanning electron microscope', which we now changed at line 135.

5) L. 201; I prefer more informative subtitle of paragraph. e.g. SEM measurement of spine morphology and densities.

We now changed the text accordingly at line 203.

6) Figure 5: Can you indicate the difference in species by color? e.g. T. sacculifer is bluegreen colors, G. ruber is red-orange colors. Describe the meanings of shown lines in this caption. Further, could you indicate which data were measured by EPAM or LA-ICP-MS?

Thank you for the suggestion, we now changed figure 5 and its caption accordingly.

7) 4.1 Organic Linings: Authors could show that organic lining were poor with Na. Why EPMA Na mapping never show OM as low concentration bands?

As explained in line 155, organic linings in these species only account for a very limited part of the total shell weight, 0.4-0.7% of the total shell. But the carbonate deposited at or close to the organic lining could even be relatively high in Na. Here we cannot observe the linings as such using EPMA imaging. We now added an extra sentence about this in line 280.

8) Figure 6: It is not appropriate to show "?" here. Describe possible explanation for "Spine Na/Ca" and "Outliers" in the plot Instead of indicating "?".

We changed the figure (6) and caption and agree that this '?' might be confusing.

9) L. 331: This is important consideration because the measured results are variable at part by part by such partial measuement method as LA-ICP-MS, SIMS, EPMA and more. The authors will mention the importance about the choice of measurement portion on the test.

We now added some extra information on the measured portion of the test in line 96

10) Figure 7: It is good useful compilation figure of Na/Ca understanding. Could you explain all indicated knowledge of these in the main text? It would be fit the paragraph started from L333 or around the sentence.

We now added some extra text at line 337.

11) L. 355: Some figure and/or previous study should be referred after "function of salinity".

Thank you, we now changed the text accordingly.

12) Figure 10: Explanations about the lines in the fig. 10a is necessary in the figure caption.

Thank you, we now changed the text accordingly.

Please also note the supplement to this comment:
https://www.biogeosciences-discuss.net/bg-2018-454/bg-2018-454-AC1-supplement.pdf

[Figure]

**Fig. 1.** Figure R1

**Supplement:**

[revised manuscript text omitted]

---

## Author Comment (AC2) · 16 Jan 2019

Dear editor and Prof. dr. Ralf Schiebel,

Thank you very much for the useful comments, additions and suggestions on our manuscript. We have changed and/or answered these comments step-by-step in the text below, explaining why we agree or respectfully disagree. Our answers are directly below the individual comments.

Also on behalf of the other authors,

Respectfully yours,

Eveline Mezger

ralf.schiebel@mpic.de It is a pleasure to read the considerate and well-written paper of Mezger and coauthors on "Planktonic foraminiferal spine versus shell carbonate Na incorporation in relation to salinity". In addition to the review of Takashi Toyofuku's, I only have two more comprehensive points to be considered by the authors, as well as some small points to be taken into consideration before publication of the paper.

1) It would be interesting and important to know, which of the morphotypes of T. sacculifer, and which morpho- and genotypes of G. ruber are analyzed here and are represented by the data. Since some of the different types have different ecological demands, the new Na/Ca possibly represent very specific ecological conditions, which in turn may be reconstructed when using Na/Ca as a proxy of past conditions. The data plotted in Figure 4 may even show more structure when being plotted separately for different morphotypes? Also, "variability within one species" (line 256) may turn out to make sense for different morphotypes (also lines 258-259) ?!

Thank you for this suggestion. We are aware that morphotypes might be reflected by elemental composition of the species (e.g. for G. ruber: Steinke et al., 2005), mostly because of the different depth habitats these morphospecies live in. At line 256 we refer to the internal shell Na variability, which seems similar for all specimens observed here (namely higher Na in spines, lower in the shell). Apparently, this internal variability does not differ between morphotypes. For the shallow-collected (surface water, multi-nets up to ~50 m depth), we were not able to differentiate between morphotypes, as the morphological traits for both species are mainly, and more clearly, pronounced in terminal reproductive stages. In the upper part of the water column, the G. ruber sensu stricto morphotype is known to dominate, whereas at great depths, the sensu lato is the dominant species (Wang, 200). Therefore, we believe for the shallow water-collected specimens that the impact of different morphospecies is rather limited. As we agree

that this is an important consideration for the development of a salinity proxy, we added your suggestion at line 257, also referring to Schiebel and Hemleben, 2017.

2) In section 3.2, Scanning electron microscopy, the authors may want to consider that there are two different types of spines both in G. ruber and T. sacculifer, one being round and the other being triangular (see, e.g., Schiebel and Hemleben, 2017). The very different spine widths reported in lines 205 to 208 may result from the two different types of round (thinner) and triangular (thicker) spines and may hence represent two groups of data in each of the species, and, even more importantly, Na/Ca may vary between the two types of spines.

Thank you for this useful comment. So far, based on SEM imaging and EPMA imaging/polishing of the shells, it looks like all spines end circular rather than triangular. Based on you comment, this implies that our conclusions can not necessarily be translated to all spine morphologies. We now added this suggestion to the discussion at line 316, also referring to Schiebel and Hemleben, 2017.

Minor points:

3) Line 22: better refer to carbonate, not calcite, because of other carbonate species like vaterite (Jacob et al., 2017); Thank you, we now changed the text accordingly.

4) in the entire manuscript Line 23: better use "taphonomic alteration" than "taponomy" (also line 79)

Thank you, we now changed the text accordingly.

5) Line 56: bivalves have no spines (Zhao et al. 2017), and I cannot see any connection here

Thank you, we now changed the text accordingly and deleted the reference

6) Line 108: the "Whole shell" is called "test" in foraminifera; why do the authors avoid the term in the entire manuscript?

Apparently, journals nowadays apparently prefer using 'shell', which I have also adopted for my other publications. Furthermore, the use of the word 'test' might be confusing, as people that are less into the material/foraminifera vocabulary might be looking for 'shell' rather than test.

7) Line 171: s-pecimen; Line 179: ruber italic; Lines 197-198: "in mixing signals between spine base and shell carbonate", possibly resulting from the resolution of measurements; Line 208: "Salinity correlates negatively with spine width", may be turned around to keep the right order; Figure 5: please indicate which data refer to entire test and which to shell (-only) measurments: One comma too many in the caption. Line 355: better: "When spines fully account for. . ." Line 367: better "up to 2-3 mm long;..." Line 374: (Figs. 5 and 8), check in the entire manuscript Line 378: . . . sacculifer); open parentheses Line 395: Na concentration?

Thank you for these comments, we now changed the text accordingly.

Please also note the supplement to this comment:
https://www.biogeosciences-discuss.net/bg-2018-454/bg-2018-454-AC2-
supplement.pdf

**Supplement:**

[revised manuscript text omitted]

---

## Referee Comment (RC3) · Toyofuku (Referee) · 17 Jan 2019

I have read the new revision. Every reply with my questions was reasonable and the contents are brushed up. I am honored if my comment was helpful. It is good job to publish on BG.

———————————————————

---

## Referee Comment (RC4) · Schiebel (Referee) · 21 Jan 2019

Further to the author's comments on my earlier review (included below), two points may be added. (1) It would always make sense to work on the highest systematic level possible, i.e. genotypes and morphotypes, if possible. (2) Both round and triangular spines differ in cross section at their base and are round at the top. It would hence be the base to look at for the respective difference in spine type. Both of the comments may be taken into consideration for future analyses to again improve the level of the scientific approach.

Earlier review:

It is a pleasure to read the considerate and well-written paper of Mezger and coauthors

on "Planktonic foraminiferal spine versus shell carbonate Na incorporation in relation to salinity". In addition to the review of Takashi Toyofuku's, I only have two more comprehensive points to be considered by the authors, as well as some small points to be taken into consideration before publication of the paper.

It would be interesting and important to know, which of the morphotypes of T. sacculifer, and which morpho- and genotypes of G. ruber are analyzed here and are represented by the data. Since some of the different types have different ecological demands, the new Na/Ca possibly represent very specific ecological conditions, which in turn may be reconstructed when using Na/Ca as a proxy of past conditions. The data plotted in Figure 4 may even show more structure when being plotted separately for different morphotypes? Also, "variability within one species" (line 256) may turn out to make sense for different morphotypes (also lines 258-259) ?!

In section 3.2, Scanning electron microscopy, the authors may want to consider that there are two different types of spines both in G. ruber and T. sacculifer, one being round and the other being triangular (see, e.g., Schiebel and Hemleben, 2017). The very different spine widths reported in lines 205 to 208 may result from the two different types of round (thinner) and triangular (thicker) spines, and may hence represent two groups of data in each of the species, and, even more importantly, Na/Ca may vary between the two types of spines.

Minor points: Line 22: better refer to carbonate, not calcite, because of other carbonate species like vaterite (Jacob et al., 2017); in the entire manuscript Line 23: better use "taphonomic alteration" than "taponomy" (also line 79) Line 56: bivalves have no spines (Zhao et al. 2017), and I cannot see any connection here Line 108: the "Whole shell" is called "test" in foraminifera; why do the authors avoid the term in the entire manuscript? Line 171: s-pecimen Line 179: ruber italic Lines 197-198: "in mixing signals between spine base and shell carbonate", possibly resulting from the resolution of measurements Line 208: "Salinity correlates negatively with spine width", may be turned around to keep the right order Figure 5: please indicate which data refer to entire test and which to shell (-only) measurments: One comma too many in the caption. Line 355: better: "When spines fully account for. . ." Line 367: better "up to 2-3 mm long ..." Line 374: (Figs. 5 and 8), check in the entire manuscript Line 378: . . . sacculifer); open parentheses Line 395: Na concentration?

---

## Author Comment (AC3) · 23 Jan 2019

Dear Prof. Schiebel,

thank you once more for your constructive comments! We can only agree with your suggestions and hope that our manuscript will show the urgency to study the relation between geochemistry and geno-/morphotypes in more detail. Secondly, we looked at the spine bases only, i.e. the place where the triangular or circular shape of the spine is most clearly visible. By including a reference to your and Chr. Hemleben's book, we show that the correlation between El/Ca and spine (bases) may vary with spine morphology.

Sincerely,

[Figure]

Eveline Mezger

Comments Prof Schiebel: "Further to the author's comments on my earlier review (included below), two points may be added. (1) It would always make sense to work on the highest systematic level possible, i.e. genotypes and morphotypes, if possible. (2) Both round and triangular spines differ in cross section at their base and are round at the top. It would hence be the base to look at for the respective difference in spine type. Both of the comments may be taken into consideration for future analyses to again improve the level of the scientific approach."

---

## Author Comment (AC4) · 15 Feb 2019

Response to RC3 of dr. Takashi Toyofuku:

"I have read the new revision. Every reply with my questions was reasonable and the contents are brushed up. I am honored if my comment was helpful. It is good job to publish on BG."

Dear dr. Takashi Toyofuku,

Thank you very much for your kind words. We believe that our manuscript improved significantly thanks to your comments and suggestions.

Also on behalf of the other authors,

[Figure]

Respectfully yours,

Eveline Mezger

---

## Author Response (AR1)

Dear associate editor Prof. Hiroshi Kitazato,

Thank you very much for your kind words, useful comments and suggestions for future research. Herewith we send you the revised version of the manuscript.

Also on behalf of the other authors,

Respectfully yours,

Eveline Mezger

[revised manuscript text omitted]